# FEAT: Free energy Estimators with Adaptive Transport

**Yuanqi Du**[2][*] **& Jiajun He**[1][*], **Francisco Vargas**[3], **Yuanqing Wang**[4],
**Carla P. Gomes**[2], **José Miguel Hernández-Lobato**[1], **Eric Vanden-Eijnden**[4,5]

[1]University of Cambridge, [2]Cornell University, [3]Xaira Therapeutics,
[4]ML Lab, Capital Fund Management, [5]Courant Institute of Mathematical Sciences, NYU

## Abstract

We present *Free energy Estimators with Adaptive Transport* (FEAT), a novel framework for free energy estimation—a critical challenge across scientific domains. FEAT leverages learned transports implemented via stochastic interpolants and provides consistent, minimum-variance estimators based on escorted Jarzynski equality and controlled Crooks theorem, alongside variational upper and lower bounds on free energy differences. Unifying equilibrium and non-equilibrium methods under a single theoretical framework, FEAT establishes a principled foundation for neural free energy calculations. Experimental validation on toy examples, molecular simulations, and quantum field theory demonstrates promising improvements over existing learning-based methods. Our PyTorch implementation is available at `https://github.com/jiajunhe98/FEAT`.

## 1 Introduction

Estimating free energy is fundamental across machine learning (appearing as normalization factors and the model evidence), statistical mechanics (partition functions), chemistry, and biology [Chipot and Pohorille, 2007, Lelièvre et al., 2010, Tuckerman, 2023]. The free energy is expressed as:

$$F = -k_B T \log Z, \qquad Z = \int_\Omega \exp(-\beta U(x)) \mathrm{d}x \tag{1}$$

where $\Omega \subseteq \mathbb{R}^d$, $U : \Omega \to \mathbb{R}$ is the energy function, assumed to be such that $Z < \infty$, and $\beta = 1/k_B T$ combines the Boltzmann constant $k_B$ and temperature $T$.

Rather than calculating $F$ directly, one typically estimates the free energy difference between systems (or states) $S_a$ and $S_b$ with energies $U_a$ and $U_b$, which is essential for biological conformational changes, ligand-macromolecule binding, and chemical reaction mechanisms [Wang et al., 2015]:

$$\Delta F = F_b - F_a = -k_B T \log(Z_b/Z_a) \tag{2}$$

This computational challenge has driven numerous approaches. Zwanzig [1954] reformulated the problem as *importance sampling*, where one system serves as the proposal, enabling free energy difference estimation via Monte Carlo sampling. This free energy perturbation (FEP) method, however, suffers from high variance when the energies $U_a$ and $U_b$ of systems $S_a$ and $S_b$ differ significantly, particularly in high-dimensional spaces.

To mitigate this issue, targeted FEP [Jarzynski, 2002] learns an invertible mapping between two distributions to increase their overlap. From a complementary angle, Bennett [1976] generalized

---

[*]The authors contributed equally to this work. The order is randomly assigned and randomly reshuffled in different version of the paper to reflect this equal contribution. Corresponding to <jh2383@cam.ac.uk>, <yuanqidu@cs.cornell.edu>, and <eve2@cims.nyu.edu>.

39th Conference on Neural Information Processing Systems (NeurIPS 2025).

FEP to create a *minimum-variance free energy estimator*—the Bennett acceptance ratio (BAR) method. However, BAR still requires sufficient distribution overlap. Shirts and Chodera [2008] addressed this limitation with the multistate Bennett acceptance ratio (MBAR), introducing multiple intermediate system to improve overlap. Thermodynamic integration [TI, Kirkwood, 1935] takes a different approach by constructing a *continuous path* of systems $S_t$ with energies $U_t$, with $t \in [0, 1]$, connecting $S_a$ and $S_b$. The free energy difference is defined by integrating instantaneous energy differences along this path using samples from each intermediate systems $S_t$—effectively performing infinitely many consecutive FEPs between infinitesimally close distributions.

The methods above rely on *equilibrium* averages, requiring exact samples drawn from the distributions of each considered state. Jarzynski equality [Jarzynski, 1997] offered a breakthrough alternative based on *non-equilibrium* trajectories. Similar to TI, methods based on this equality utilize a path of intermediate systems $S_t$, but only require exact samples from one endpoint, $S_a$ or $S_b$, allowing the law of the transported samples to deviate from the law of the intermediate systems. The escorted Jarzynski equality [Vaikuntanathan and Jarzynski, 2008] further refined this approach by introducing additional control that reduce estimator variance by constraining these deviations.

Recent advances have leveraged the capacity of neural networks to approximate high-dimensional distributions for improved free energy estimation. In Targeted FEP approaches, researchers have developed invertible maps using normalizing flows [Wirnsberger et al., 2020] and flow matching [Zhao and Wang, 2023, Erdogan et al., 2025]. In the context of TI, Máté et al. [2024a,b] introduced energy-parameterized diffusion [Song et al., 2021] and stochastic interpolant models [Albergo et al., 2023] to capture energy interpolants between endpoint distributions.

Despite recent advances, non-equilibrium approaches and their connections to other methods remain under-explored in the deep learning landscape for free energy estimation. Our work addresses this gap by introducing *Free energy Estimators with Adaptive Transport* (FEAT). Using samples from both endpoints, FEAT constructs non-equilibrium transport via stochastic interpolants. This learned transport is then leveraged through the escorted Jarzynski equality and Crooks theorem to obtain both variational bounds and a consistent, minimum-variance estimator for free energy differences.

FEAT capitalizes on the key advantage of non-equilibrium transport: eliminating the need for exact samples at intermediate distributions, thereby enabling more efficient computation. It facilitates faster numerical simulations without costly divergence evaluations while demonstrating enhanced robustness to discretization and network learning errors. Notably, our framework subsumes existing equilibrium-based methods as special cases, revealing a larger design space with greater flexibility and performance potential. Experimental results confirm that FEAT significantly outperforms leading baselines, including targeted FEP and neural thermodynamic integration.

## 2 Background and Motivation

Here we summarize key free energy estimators most relevant to our approach. While this material is well-established in the computational physics literature, it may be less familiar to machine learning audiences. For the reader's convenience, we provide a more comprehensive discussion and derivations of these results in Appendix B. For simplicity, hereafter, we set the combination of Boltzmann constant and temperature $k_B T = \beta^{-1} = 1$ to absorb it into the definitions of the potential and free energy.

### 2.1 Free energy perturbation and Bennett acceptance ratio

*Free energy perturbation* [FEP, Zwanzig, 1954] estimates free energy differences between systems $S_a$ and $S_b$ through importance sampling, using samples from one system as proposals for the other. This gives the following expression for the free energy difference $\Delta F$:

$$\Delta F = -\log(Z_b/Z_a) = -\log \mathbb{E}_a \big[ \exp(U_a - U_b) \big], \tag{3}$$

where we use $\mathbb{E}_a$ to denote the expectation with respect to the equilibrium distribution $\mu_a(\mathrm{d}x) = Z_a^{-1} e^{-U_a(x)} \mathrm{d}x$ of system $S_a$.

*Bennett acceptance ratio* [BAR, Bennett, 1976] extends FEP with a minimal variance estimator (specifically, minimal relative mean squared error):

$$\Delta F = -\log(Z_b/Z_a) = -\log \frac{\mathbb{E}_a \big[ \phi(U_b - U_a - C) \big]}{\mathbb{E}_b \big[ \phi(U_a - U_b + C) \big]} + C \tag{4}$$

where $\mathbb{E}_a$ and $\mathbb{E}_b$ denote expectations with respect to the equilibrium distributions of systems $S_a$ and $S_b$, and $\phi(x)$ is any function satisfying $\phi(x)/\phi(-x) = e^{-x}$: the usual choice is the Fermi function $\phi(x) = 1/(1 + e^x)$. The constant $C$ can be optimized to minimizes the variance of the estimator: this gives $C = \Delta F$. Since $\Delta F$ is unknown initially, in practice it is determined iteratively by updating $C$ to $\Delta \hat{F}$, where $\Delta \hat{F}$ is computed from Equation (4) in the previous iteration.

Both FEP and BAR rely on importance sampling and become ineffective when the distributions of systems $S_a$ and $S_b$ have minimal overlap. *Targeted FEP* [Jarzynski, 2002] addresses this limitation by designing an invertible transformation $T$ that maps samples between states. The free energy difference is then calculated through importance sampling with change of variable:

$$\Delta F = -\log \mathbb{E}_a\big[\exp(-\Phi)\big], \quad \Phi(x) = U_b(T(x)) - U_a(x) - \log|\nabla T(x)| \tag{5}$$

where $\nabla T(x)$ denotes the Jacobian matrix of the map $T$ and $|\nabla T(x)|$ its determinant. This approach naturally integrates with neural density estimators. Recent works have implemented this transformation using normalizing flows [Wirnsberger et al., 2020], computing the Jacobian via an invertible network, and flow matching [Lipman et al., 2023, Zhao and Wang, 2023, Erdogan et al., 2025], computing the Jacobian via the instantaneous change of variable formula.

## 2.2 Jarzynski equality, Crooks fluctuation theorem, and their escorted variations

Let $U_t$ be a smooth energy interpolating between $U_{t=0} = U_a$ and $U_{t=1} = U_b$, and consider the stochastic process governed by the Langevin equation over this evolving potential:

$$\mathrm{d}X_t = -\sigma_t^2 \nabla U_t(X_t)\mathrm{d}t + \sqrt{2}\sigma_t \overrightarrow{\mathrm{d}B}_t, \quad X_0 \sim \mu_a, \tag{6}$$

where $\sigma_t \geq 0$ is the volatility , $X_0 \sim \mu_a$ indicates that $X_0$ is sampled from the distribution $\mu_a(\mathrm{d}x) = Z_a^{-1} e^{-U_a(x)}\mathrm{d}x$, and the arrow over the Brownian motion $B_t$ indicates that this equation must be solved forward in time. Because $U_t$ is time-dependent, the law of $X_t$ is not the Gibbs distribution associated with $U_t$. Yet, *Jarzynski equality* [Jarzynski, 1997] shows that we can correct for these non-equilibrium effects and relate the (equilibrium) free energy difference to the work $W$:

$$\Delta F = -\log \mathbb{E}_{\overrightarrow{\mathbb{P}}}[\exp(-W)], \quad W(X) = \int_0^1 \partial_t U_t(X_t)\mathrm{d}t \tag{7}$$

where $\mathbb{E}_{\overrightarrow{\mathbb{P}}}$ denotes expectation over the path measure $\overrightarrow{\mathbb{P}}$ of the solutions to Equation (6). Note that, if we set $\sigma_t = 0$ in Equation (6), the samples do not move so that we have $X_t = X_0 \sim \mu_a$ and $W_{a\to b}(X) = \int_0^1 \partial_t U_t(X_t)\mathrm{d}t = U_b(X_0) - U_a(X_0)$, so that Equation (7) reduces to Equation (3). The interest of Equation (7) is that it also works when $\sigma_t > 0$.

We can also express and interpret Jarzynski equality through *Crooks fluctuation theorem* [CFT, Crooks, 1999]. Specifically, consider the following backward SDEs with path measure $\overleftarrow{\mathbb{P}}$:

$$\mathrm{d}X_t = \sigma_t^2 \nabla U_t(X_t)\mathrm{d}t + \sqrt{2}\sigma_t \overleftarrow{\mathrm{d}B}_t, \quad X_1 \sim \mu_b, \tag{8}$$

where $X_1 \sim \mu_b$ indicates that $X_1$ is sampled from the distribution $\mu_b(\mathrm{d}x) = Z_b^{-1} e^{-U_b(x)}\mathrm{d}x$ and the arrow over the Brownian motion $B_t$ indicates that this equation must be solved backward in time. Assuming that $\sigma_t > 0$, the Radon-Nikodym derivative between the path measure of forward and the backward processes solutions to Equation (6) and Equation (8), respectively, can be expressed in terms of the free energy difference and the work as

$$\frac{\mathrm{d}\overleftarrow{\mathbb{P}}}{\mathrm{d}\overrightarrow{\mathbb{P}}}(X) = \exp(-W(X) + \Delta F) \tag{9}$$

Jarzynski equality can be recovered from this expression by noting that its expectation over $\overrightarrow{\mathbb{P}}$ is 1.

Vaikuntanathan and Jarzynski [2008] add a control term $v_t$ to the drift in Equation (6), whose aim is to better align the law of $X_t$ with the Gibbs distribution associated with $U_t$:

$$\mathrm{d}X_t = -\sigma_t^2 \nabla U_t(X_t)\mathrm{d}t + v_t(X_t)\mathrm{d}t + \sqrt{2}\sigma_t \overrightarrow{\mathrm{d}B}_t, \quad X_0 \sim \mu_a \tag{10}$$

Let $\overrightarrow{\mathbb{P}}^v$ be the path measure of the solution to this SDE and define the *generalized work* $W^v$ as:

$$W^v(X) = \int_0^1 \left(-\nabla \cdot v_t(X_t) + \nabla U_t(X_t) \cdot v_t(X_t) + \partial_t U_t(X_t)\right)\mathrm{d}t \tag{11}$$

where $\nabla\cdot$ represents the divergence operator, i.e., trace of Jacobian.

*Escorted Jarzynski equality* expresses the free energy difference in terms this generalized work as:

$$\Delta F = -\log \mathbb{E}_{\overrightarrow{\mathbb{P}}^v}[\exp(-W^v)]. \tag{12}$$

This expression remains valid if we use $\sigma_t = 0$ in Equation (10) and it reduces to the ODE

$$\mathrm{d}X_t = v_t(X_t)\mathrm{d}t, \quad X_0 \sim \mu_a \qquad (\sigma_t = 0) \tag{13}$$

By chain rule, we have $\nabla U_t(X_t) \cdot v_t(X_t) + \partial_t U_t(X_t) = (\mathrm{d}/\mathrm{d}t)U_t(X_t)$, and hence the generalized work in Equation (12) becomes $W^v(X) = -\int_0^1 \nabla \cdot v_t(X_t)\mathrm{d}t + U_b(X_1) - U_a(X_0)$. In this case, this expression becomes an implementation of Equation (5) in which we construct the map $T$ via solution of the ODE (13) by setting $T(X_0) = X_{t=1}$ and hence $\log|\nabla T| = \int_0^1 \nabla \cdot v_t(X_t)\mathrm{d}t$. This ODE-based mapping is also known as the continuous normalizing flow [CNF, Chen et al., 2018]. We will come back to this connection in Section 3.5.

It can also be shown [Lelièvre et al., 2010, Heng et al., 2021, Arbel et al., 2021, Vargas et al., 2024, Albergo and Vanden-Eijnden, 2024] that the law of the solution to Equation (6) is precisely $\mu_t(\mathrm{d}x) = Z_t^{-1}e^{-U_t(x)}\mathrm{d}x$ if and only if $v_t(x)$ satisfies

$$-\nabla \cdot v_t(x) + \nabla U_t(x) \cdot v_t(x) + \partial_t U_t(x) = \partial_t F_t, \qquad F_t = -\log Z_t. \tag{14}$$

In this case, the generalized work defined in Equation (11) is determinitic and given by $W^v(X) = \int_0^1 \partial_t F_t\mathrm{d}t = \Delta F$, and hence the escorted Jarzynski equality becomes a practical way to implement *thermodynamic integration* (TI). We elaborate on this connection in Section 3.5.

When $\sigma_t > 0$, we can also establish the *controlled Crooks fluctuation theorem* [Vargas et al., 2024] by considering the backward SDE:

$$\mathrm{d}X_t = \sigma_t^2 \nabla U_t(X_t)\mathrm{d}t + v_t(X_t)\mathrm{d}t + \sqrt{2}\sigma_t\overleftarrow{\mathrm{d}B_t}, \quad X_1 \sim \mu_b \tag{15}$$

Denoting the path measure of the solution to this SDE as $\overleftarrow{\mathbb{P}}^v$, we have:

$$\frac{\mathrm{d}\overleftarrow{\mathbb{P}}^v}{\mathrm{d}\overrightarrow{\mathbb{P}}^v}(X) = \exp(-W^v(X) + \Delta F) \tag{16}$$

The expectation of this expression over $\overrightarrow{\mathbb{P}}^v$ is 1, which recovers Equation (12).

## 3 Methods

To leverage the escorted Jarzynski equality effectively, we need a control term $v_t$ that enables Equation (10) to transport samples from $S_a$ to $S_b$ accurately. Recent neural samplers [Vargas et al., 2024, Albergo and Vanden-Eijnden, 2024] approach this by first defining an energy interpolant (e.g., $U_t = (1-t)U_a + tU_b$), then optimizing a neural network to learn $v_t$ in Equation (10). These methods address the challenging scenario where only samples from one endpoint are available, making their performance sensitive to the choice of interpolating energy $U_t$ [Syed et al., 2022, Máté and Fleuret, 2023, Phillips et al., 2024] and requiring Langevin dynamics trajectory simulation during training.

Our setting is different: like BAR and other established methods, we assume that we have access to samples from both systems $S_a$ and $S_b$, and our goal is solely to compute the free energy difference between these endpoints. This simplifies matters and renders the specific choice of energy interpolant $U_t$ less critical. In particular, it allows us to leverage stochastic interpolants framework [Albergo and Vanden-Eijnden, 2023, Albergo et al., 2023] to develop an efficient and scalable method for simultaneously learning the transport between the two marginal distributions and the associated energy function $U_t$. This learning-based strategy is explained next and summarized in Algorithm 1.

### 3.1 Learning a Transport with Stochastic Interpolants

Given samples from systems $S_a$ and $S_b$, stochastic interpolants [Albergo and Vanden-Eijnden, 2023, Albergo et al., 2023] provide a simple and efficient way to effectively learn a transport between these states in the form of Equation (10). We first define an interpolant between endpoint samples:

$$I_t = \alpha_t x_a + \beta_t x_b + \gamma_t \epsilon, \quad x_a \sim \mu_a, \quad x_b \sim \mu_b, \quad \epsilon \sim \mathcal{N}(0, \mathrm{Id}) \tag{17}$$

where $\alpha_0 = 1, \alpha_1 = 0$; $\beta_0 = 0, \beta_1 = 1$; and $\gamma_0 = \gamma_1 = 0$ ensure proper boundary conditions: $I_{t=0} = x_a$ and $I_{t=1} = x_b$. From the results of Albergo et al. [2023], we know that the law of $I_t$ is, at any time $t \in [0, 1]$, the same as the law of the solution to Equation (10) if we use

$$v_t(x) = \mathbb{E}[\dot{I}_t | I_t = x], \qquad \nabla U_t(x) = \gamma_t^{-1} \mathbb{E}[\epsilon | I_t = x] \tag{18}$$

where the dot denotes the time derivative and $\mathbb{E}[\cdot | I_t = x]$ denotes expectation over the law of $I_t$ conditional on $I_t = x$. Using the $L^2$ formulation of the conditional expectation, we can write objective functions for the function $v_t$ and $\nabla U_t$ defined in Equation (18); if we parametrize these functions as neural networks $v_t^\psi(x)$ and $U_t^\theta(x)$, depending on both $t$ and $x$, this leads to the losses:

$$\mathcal{L}_v(\psi) = \mathbb{E}_{t \sim \mathcal{U}(0,1)} \mathbb{E}_{x_a, x_b, \epsilon} \left[ \lambda_t | v_t^\psi(I_t) - \dot{I}_t |^2 \right] \tag{19}$$

$$\mathcal{L}_U^{\mathrm{DSM}}(\theta) = \mathbb{E}_{t \sim \mathcal{U}(0,1)} \mathbb{E}_{x_a, x_b, \epsilon} \left[ \eta_t | \nabla U_t^\theta(I_t) - \gamma_t^{-1} \epsilon |^2 \right] \tag{20}$$

where DSM stands for denoising score matching [Vincent, 2011], and $\lambda_t$ and $\eta_t$ are weighting functions to balance optimization across different times. In practice, $\lambda_t = 1$ and $\eta_t = \gamma_t$ work well.

Once we have learned $v_t^\psi$ and $\nabla U_t^\theta$ we can use these functions in any of the estimators presented in Section 2 via computation of the generalized work in Equation (11) or the forward-backward Radon-Nikodym derivative (FB RND) in Equation (16). Note that, Equation (11) requires $\partial_t U_t$, necessitating an energy-parameterized network $U_t^\theta$, which is known to be difficult to learn via score matching [Zhang et al., 2022]. In contrast, the FB RND form only requires the score function, allowing direct parameterization of $\nabla U_t^\theta$ as a score network. In fact, using FB RND formulation also offers additional benefits, which we will discuss in Section 3.4.

We stress that *the estimators presented in Section 2 are asymptotically unbiased even if we use them with functions $v_t^\psi$ and $U_t^\theta$ that have been learned imperfectly, as long as we satisfy the boundary conditions $U_{t=0}^\theta = U_a$ and $U_{t=1}^\theta = U_b$*. In practice, however, imposing these boundary conditions puts constraints on the neural network used to parameterize $U_t^\theta$ or $\nabla U_t^\theta$, which may limit its expressivity and impede training convergence. For these reasons, in FEAT we choose to not impose the boundary conditions by the parameterization design, but rather learn $U_t^\theta$ at the endpoints as well. To this end, we enhance the denoising score matching loss in Equation (20) with target score matching [TSM, De Bortoli et al., 2024]:

$$\mathcal{L}_U^{\mathrm{TSM},0}(\theta) = \mathbb{E}_{t \sim \mathcal{U}(0,0.5)} \mathbb{E}_{x_a, x_b, \epsilon} \left[ |\nabla U_t^\theta(I_t) - \alpha_t^{-1} \nabla U_a(x_a)|^2 \right] \tag{21}$$

$$\mathcal{L}_U^{\mathrm{TSM},1}(\theta) = \mathbb{E}_{t \sim \mathcal{U}(0.5,1)} \mathbb{E}_{x_a, x_b, \epsilon} \left[ |\nabla U_t^\theta(I_t) - \beta_t^{-1} \nabla U_b(x_b)|^2 \right] \tag{22}$$

This objective formulation was introduced by Máté et al. [2024b] to improve energy estimation accuracy in neural thermodynamic integration (TI). Importantly, TSM does not increase target energy evaluation costs, as the gradients $\nabla U_a(x_a)$ and $\nabla U_b(x_b)$ can be precomputed and stored during molecular dynamics (MD) simulations alongside collected samples.

Unfortunately, even though TSM largely increases the accuracy of the boundary conditions, error still exists due to imperfect learning. Next, we discuss how to generalize the escorted Jarzynski estimators in Section 2 to account for the mismatch between $U_t^\theta$ and $U_a$ and $U_b$ at the endpoints.

## 3.2 Estimating the Free Energy Difference with Escorted Jarzynski Equality

Suppose that we have learned a transport with stochastic interpolants for which the boundary conditions are not necessarily satisfied, meaning $U_0^\theta \neq U_a$ and $U_1^\theta \neq U_b$. This boundary mismatch requires a correction term, as specified by the following result:

**Corollary 3.1** (Escorted Jarzynski Equality with imperfect boundary conditions). *Given $v_t^\psi$ and $U_t^\theta$, consider the forward and backward SDEs:*

$$\mathrm{d}X_t = -\sigma_t^2 \nabla U_t^\theta(X_t) \mathrm{d}t + v_t^\psi(X_t) \mathrm{d}t + \sqrt{2} \sigma_t \overrightarrow{\mathrm{d}B_t}, \quad X_0 \sim \mu_a, \tag{23}$$

$$\mathrm{d}X_t = \sigma_t^2 \nabla U_t^\theta(X_t) \mathrm{d}t + v_t^\psi(X_t) \mathrm{d}t + \sqrt{2} \sigma_t \overleftarrow{\mathrm{d}B_t}, \quad X_1 \sim \mu_b, \tag{24}$$

*where $\sigma_t \geq 0$ and $\mu_a$ and $\mu_b$ denote the distributions associated with the energies $U_a$ and $U_b$, respectively. Define also the "corrected generalized work":*

$$\widetilde{W}^v(X) = \int_0^1 \left( -\nabla \cdot v_t^\psi(X_t) + \nabla U_t^\theta(X_t) \cdot v_t^\psi(X_t) + \partial_t U_t^\theta(X_t) \right) \mathrm{d}t + \underbrace{\log \frac{\exp(-U_a(X_0)) \exp(-U_1^\theta(X_1))}{\exp(-U_b(X_1)) \exp(-U_0^\theta(X_0))}}_{\text{correction term}} \tag{25}$$

*Using the generalized work with correction, we have the same escorted Jarzynski equality as before:*

$$\Delta F = -\log \mathbb{E}_{\overrightarrow{\mathbb{P}}^v}[\exp(-\widetilde{W}^v)] = \log \mathbb{E}_{\overleftarrow{\mathbb{P}}^v}[\exp(\widetilde{W}^v)] \tag{26}$$

*where $\overrightarrow{\mathbb{P}}^v$ and $\overleftarrow{\mathbb{P}}^v$ are the path measures over the solutions to Equations* (23) *and* (24)*, respectively.*

The proof of this proposition is given in Appendix C.3. Note that the corrected generalized work in Equation (25) coincides with the generalized work in Equation (11) if $U_0^\theta = U_a$ and $U_1^\theta = U_b$, but we stress again the proposition remains valid even if these equalities do not hold.

The escorted Jarzynski equality in Equation (26) can be used to estimate the free energy difference: Denoting by $\overrightarrow{X}^{(1:N)} \sim \overrightarrow{\mathbb{P}}^v$ and $\overleftarrow{X}^{(1:N)} \sim \overleftarrow{\mathbb{P}}^v$ $N$ independent realizations of the solutions to the forward Equation (23) and the backward Equation (24), respectively, we have

$$\Delta F \approx -\log {}^1\!/_N \sum_{n=1}^N \exp(-\widetilde{W}^v(\overrightarrow{X}^{(n)})) \approx \log {}^1\!/_N \sum_{n=1}^N \exp(\widetilde{W}^v(\overleftarrow{X}^{(n)})) \tag{27}$$

and these expressions become unbiased in the limit as $N \to \infty$. These finite sample size estimators coincide with the importance-weighted autoencoder [IWAE, Burda et al., 2015], and they give us bounds on the free energy difference in expectation:

$$\mathbb{E}\left[\log {}^1\!/_N \sum_{n=1}^N \exp(\widetilde{W}^v(\overleftarrow{X}^{(n)}))\right] \leq \Delta F \leq -\mathbb{E}\left[\log {}^1\!/_N \sum_{n=1}^N \exp(-\widetilde{W}^v(\overrightarrow{X}^{(n)}))\right] \tag{28}$$

These bounds are much tighter in general than the usual variational bounds:

$$\mathbb{E}_{\overleftarrow{\mathbb{P}}^v}[\widetilde{W}^v] \leq \Delta F \leq \mathbb{E}_{\overrightarrow{\mathbb{P}}^v}[\widetilde{W}^v] \tag{29}$$

which are also known as the evidence lower and upper bounds (ELBO and EUBO) in variational inference [Blei et al., 2017, Ji and Shen, 2019, Blessing et al., 2024] and were applied to free energy estimation by Wirnsberger et al. [2020], Zhao and Wang [2023] in their targeted FEP. We prove the IWAE bounds and the variational bounds in Appendix C.1.

## 3.3 Minimizing Variance with Non-equilibrium Bennett Acceptance Ratio

Equation (27) provides two estimators for the free energy difference. While simply averaging them can reduce variance, we can achieve optimal variance reduction using an approach similar to Bennett's Acceptance Ratio [BAR, Bennett, 1976].

**Proposition 3.2** (Minimum variance non-equilibrium free energy estimator)**.** *Let $\widetilde{W}^v$ be the work terms defined in Corollary 3.1. The minimum-variance estimator is given by:*

$$\Delta F \approx \log \frac{\sum_{m=1}^N \phi(-\widetilde{W}^v(\overleftarrow{X}^{(m)}) + C)}{\sum_{n=1}^N \phi(\widetilde{W}^v(\overrightarrow{X}^{(n)}) - C)} + C, \quad \overrightarrow{X}^{(1:N)} \sim \overrightarrow{\mathbb{P}}^v, \quad \overleftarrow{X}^{(1:N)} \sim \overleftarrow{\mathbb{P}}^v \tag{30}$$

*where $\phi$ is the Fermi function $\phi(z) = 1/(1 + \exp(z))$. In addition, the minimum variance estimator is obtained with $C = \Delta F$.*

In practice, we initialize $C$ as the mean of Equation (27), then iteratively update $\Delta F$ using $C$ set to the current estimate until convergence. This estimator was originally introduced by Bennett [1976] for equilibrium averages, with non-equilibrium variants later developed by Shirts et al. [2003], Hahn and Then [2009], Minh and Chodera [2009], Vaikuntanathan and Jarzynski [2011] using work likelihood or "density of trajectory" concepts. In Appendix C.2, we provide an alternative derivation based on the Radon-Nikodym derivative between path measures.

## 3.4 Improving Accuracy and Efficiency of Free Energy Estimation with FB RND

We now turn our attention to estimators of the free energy difference using forward-backward Radon-Nikodym derivative (FB RND), which is enabled by the (controlled) Crooks fluctuation theorem. This approach allows us to address an issue we have left open: in practice, numerical integration of Equations (23) and (24) requires time discretization, introducing additional error. We demonstrate below that FB RND-based estimators yield asymptotically unbiased estimates of free energy differences, even in the presence of this discretization error.

We first describe the calculation of the "corrected generalized work" with discretized FB RND: let us discretize the time interval $[0, 1]$ into $M$ steps $t_0 = 0 < t_1 < \cdots < t_{M-1} < t_M = 1$ with step size $\Delta t$, and denote by $\mathcal{N}^+$ and $\mathcal{N}^-$ the transition kernels under Euler-Maruyama discretization for the forward and backward SDEs in Equations (23) and (24):

$$\mathcal{N}^+(X_{t_{i+1}}|X_{t_i}) = \mathcal{N}\left(X_{t_{i+1}}|X_{t_i} + v_{t_i}^\psi(X_{t_i})\Delta t - \sigma_{t_i}^2 \nabla U_{t_i}^\theta(X_{t_i})\Delta t, 2\sigma_{t_i}^2 \Delta t\right) \tag{31}$$

$$\mathcal{N}^-(X_{t_{i-1}}|X_{t_i}) = \mathcal{N}\left(X_{t_{i-1}}|X_{t_i} - v_{t_i}^\psi(X_{t_i})\Delta t - \sigma_{t_i}^2 \nabla U_{t_i}^\theta(X_{t_i})\Delta t, 2\sigma_{t_i}^2 \Delta t\right) \tag{32}$$

The "corrected generalized work" in the FB RND form can be calculated as:

$$\widetilde{W}^v(X) = \Delta F - \log \frac{\mathrm{d}\overleftarrow{\mathbb{P}}^v}{\mathrm{d}\overrightarrow{\mathbb{P}}^v}(X) \approx \log \frac{\exp(-U_a(X_0)) \prod_{i=1}^M \mathcal{N}^+(X_{t_i}|X_{t_{i-1}})}{\exp(-U_b(X_1)) \prod_{i=1}^M \mathcal{N}^-(X_{t_{i-1}}|X_{t_i})} \tag{33}$$

Even though the quantities at the right hand-sides are only approximate expressions for those at the left hand-side, they give consistent estimators for the free energy:

**Proposition 3.3** (Discretized controlled Crooks theorem with imperfect boundary conditions)**.** *Let* $\mathcal{N}^+$ *and* $\mathcal{N}^-$ *be as in Equations* (31) *and* (32)*, and define the forward/backward discretized paths:*

$$\overrightarrow{X}_{t_{i+1}} = \overrightarrow{X}_{t_i} - \sigma_{t_i}^2 \nabla U_{t_i}^\theta(\overrightarrow{X}_{t_i})\Delta t_i + v_{t_i}^\psi(\overrightarrow{X}_{t_i})\Delta t_i + \sqrt{2\Delta t_i}\sigma_{t_i} \eta_i, \qquad \overrightarrow{X}_0 \sim \mu_a, \quad (34)$$

$$\overleftarrow{X}_{t_{i-1}} = \overleftarrow{X}_{t_i} - \sigma_{t_i}^2 \nabla U_{t_i}^\theta(\overleftarrow{X}_{t_i})\Delta t_{i-1} - v_{t_i}^\psi(\overleftarrow{X}_{t_i})\Delta t_{i-1} + \sqrt{2\Delta t_{i-1}}\sigma_{t_i} \eta_i, \qquad \overleftarrow{X}_1 \sim \mu_b, \quad (35)$$

*where we assume that* $\sigma_t > 0$*,* $\Delta t_i = t_{i+1} - t_i$ *and* $\eta_i \sim N(0, Id)$*, independent for each* $i = 0, \ldots, M$*. Then, we have:*

$$\Delta F = -\log \mathbb{E}\left[\frac{\exp(-U_b(\overrightarrow{X}_1)) \prod_{i=1}^M \mathcal{N}^-(\overrightarrow{X}_{t_{i-1}}|\overrightarrow{X}_{t_i})}{\exp(-U_a(\overrightarrow{X}_0)) \prod_{i=1}^M \mathcal{N}^+(\overrightarrow{X}_{t_i}|\overrightarrow{X}_{t_{i-1}})}\right] = \log \mathbb{E}\left[\frac{\exp(-U_a(\overleftarrow{X}_0)) \prod_{i=1}^M \mathcal{N}^+(\overleftarrow{X}_{t_i}|\overleftarrow{X}_{t_{i-1}})}{\exp(-U_b(\overleftarrow{X}_1)) \prod_{i=1}^M \mathcal{N}^-(\overleftarrow{X}_{t_{i-1}}|\overleftarrow{X}_{t_i})}\right] \tag{36}$$

The proof is given in Appendix C.3. This proposition shows that the FB RND yields estimators that are **asymptotically unbiased** in the limit of infinite sample size, **even when the time-step size is finite**. Furthermore, imperfect boundary conditions do not alter the form of the FB RND; Equation (36) hold whether $U_0^\theta = U_a$, $U_1^\theta = U_b$ or $U_0^\theta \neq U_a$, $U_1^\theta \neq U_b$. This results in a more compact formulation for calculating the generalized work compared to Equation (25).

The FB RND approach also offers another advantage in terms of **computational efficiency** as it allows direct parameterization of the score $\nabla U_t^\theta$ without divergence calculations, eliminating backpropagation needs during training and sampling, thus enhancing efficiency and scalability. In contrast, the calculation based on Equation (25) requires both time derivative $\partial_t U_t^\theta$ and divergence $\nabla \cdot v_t^\psi$, complicating training and sampling.

**In summary,** to estimate the free energy difference between two systems $S_a$ and $S_b$, our approach learns stochastic interpolants using their samples. This yields forward and backward SDEs, as detailed in Equations (23) and (24), with potentially imperfect boundary conditions as discussed, which approximately transport samples between the two states. We then map samples from both states using these SDEs, and compute the "generalized work" via FB RND as in Equation (33). This computation results in a consistent, minimal-variance estimator for the free energy difference as defined in Equation (30). Since our method relies on the escorted Jarzynski equality with a learned escorting term adaptive to the data, we dub it the *Free energy Estimators with Adaptive Transport* (FEAT). An outline of FEAT is provided in Algorithm 1.

### 3.5 Limiting Cases of FEAT with Connections to Other Approaches

Our algorithm generalizes several existing approaches and reduces to them under specific conditions. We illustrate these relationships in Figure 1 and elaborate below, focusing on connections beyond the already established link to the (escorted) Jarzynski equality.

**FEP and Target FEP**. Our method generalizes targeted FEP with flow matching [Zhao and Wang, 2023] when setting $\sigma_t = 0$. Specifically, escorted Jarzynski equality becomes equivalent to targeted FEP with instantaneous change of variables [Chen et al., 2018] in this deterministic limit, even with imperfect boundary conditions as in Corollary 3.1. We show this equivalence in Appendix C.4. Taking further reduction, our method also generalizes standard FEP when both $\sigma_t = 0$ and $v_t = 0$.

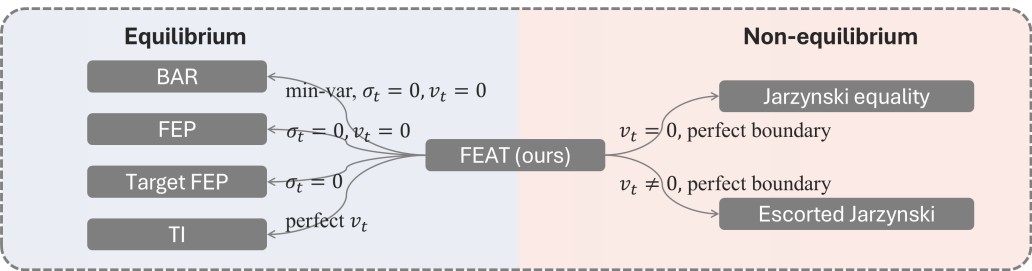

Figure 1: Connection between our proposed algorithm and other free-energy estimation approaches.

Table 1: Comparative free energy difference ($\Delta F$, unitless) estimation using targeted FEP [Zhao and Wang, 2023] and our proposed approach. MBAR results serve as reference values for LJ and ALDP systems. We report the mean and std across three runs with different seeds. (*) indicates prohibitively expensive computation due to divergence operations.

| Method | GMM (16 modes to 40 modes) | | LJ (w/o to w. LJ interaction) | | | ALDP-S | ALDP-T |
|---|---|---|---|---|---|---|---|
| | $d = 40$ | $d = 100$ | $d = 55 \times 3$ | $d = 79 \times 3$ | $d = 128 \times 3$ | $d = 22 \times 3$ | $d = 22 \times 3$ |
| Reference | 0 | 0 | 234.77 $_{\pm 0.09}$ | 357.43 $_{\pm 3.43}$ | 595.98 $_{\pm 0.58}$ | 29.43 $_{\pm 0.01}$ | -4.25 $_{\pm 0.05}$ |
| TargetFEP (FM) | 0.09 $_{\pm 0.26}$ | -17.96 $_{\pm 1.49}$ | 232.06 $_{\pm 0.03}$ | * | * | 29.47 $_{\pm 0.22}$ | -4.78 $_{\pm 0.32}$ |
| FEAT (ours) | 0.04 $_{\pm 0.04}$ | -5.34 $_{\pm 1.52}$ | 232.47 $_{\pm 0.15}$ | 356.74 $_{\pm 0.79}$ | 595.04 $_{\pm 6.52}$ | 29.38 $_{\pm 0.04}$ | -4.56 $_{\pm 0.08}$ |

In this scenario, the dynamic transport vanishes completely, and we revert to simple importance sampling using equilibrium samples from both endpoints.

**Bennett Acceptance Ratio (BAR).** Similar to FEP, our method recovers BAR when setting both $\sigma_t = 0$ and $v_t = 0$ and applying the minimum-variance estimator in Equation (30).

**Thermodynamic Integration (TI).** When $U_t^\theta$ and $v_t^\psi$ are optimally learned, our method recovers TI. Specifically, this occurs when the distribution of $X_t$ simulated from Equation (23) exactly matches the density defined by energy $U_t$. We derive this equivalence in Appendix C.5.

This connection reveals a limitation of TI: the energy function must precisely match the sample distribution. In neural TI [Máté et al., 2024a,b], the energy network must accurately capture the data density at every time step $t$, or the estimator becomes significantly biased. Our approach, based on escorted Jarzynski equality which accommodates non-equilibrium trajectories, remains effective even with imperfect learning, similar to Vargas et al. [2024], Albergo and Vanden-Eijnden [2024].

### 3.6 One-sided FEAT

In the last section, we focus on estimating the free energy *difference* between two states by learning a stochastic interpolant model. Notably, the same formulation also applies to estimating the *absolute* free energy, by choosing one of the states to be a reference distribution with known free energy, such as a Gaussian. In this case, we do not need to learn both the vector field and the score independently, as they are related in closed form [Albergo et al., 2023]. In fact, in this case, the one-sided stochastic interpolant recovers a diffusion model [Song et al., 2021]. The diffusion model learns a score network $s_t^\theta$, from which one can easily recover the vector field. We refer to this variant as *one-sided FEAT*.

To estimate the free energy difference between two arbitrary states, one can also apply one-sided FEAT to each state and then compute the difference between their estimated absolute free energies. This formulation can be viewed as an extension of DeepBAR [Ding and Zhang, 2021], which estimates absolute free energies using a normalizing flow with the BAR equation. In contrast, our approach replaces the normalizing flow with FEAT, enabling more flexible and scalable modeling. Empirically, we found DeepBAR with one-sided FEAT achieves better performance compared to using one FEAT directly bridging two states, especially on large systems. A potential reason is learning the transport from a Gaussian distribution to a complex target is simpler than learning it directly between two different complex targets.

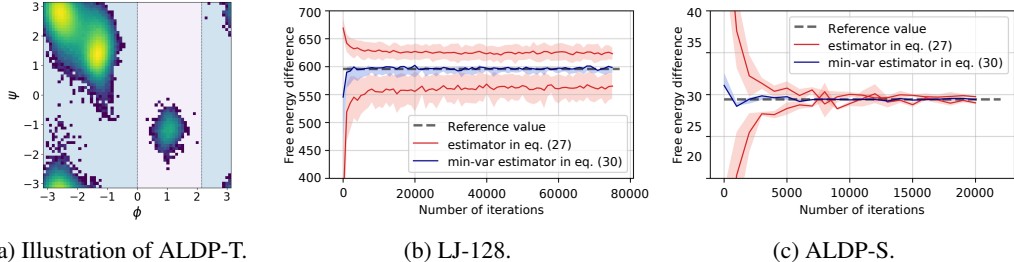

(a) Illustration of ALDP-T.      (b) LJ-128.      (c) ALDP-S.

Figure 2: (a) Two states of ALDP-T. $S_a$: $\phi \in (0, 2.15)$; $S_b$: $\phi \in [-\pi, 0] \cup [0, \pi)$; (b) (c) Estimators with eqs. (27) and (30) and their dynamics along training in LJ-128 and ALDP-S.

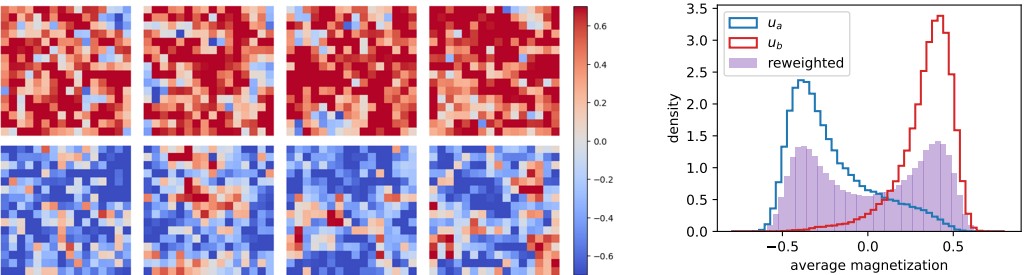

Figure 3: Eight example lattice configurations. We can see that the lattices are either mostly positive or negative.

Figure 4: Umbrella samples and reweighted histogram. We denote the two umbrellas as $u_a$ and $u_b$.

## 4 Experiments

We evaluate FEAT on a diverse range of systems, from toy examples to molecular simulations and quantum field theory. Detailed experimental setups and hyperparameters appear in Appendix F.

**Comparison with Target FEP.** We benchmark our approach against targeted FEP with flow matching [Zhao and Wang, 2023] across four systems of increasing complexity: (1) Gaussian mixtures, (2) Lennard-Jones potentials with varying parameters, (3) alanine dipeptide in vacuum vs. implicit solvent (ALDP-S), and (4) alanine dipeptide in two metastable phases (ALDP-T, illustrated in Figure 2a). For fair comparison, all methods use identical model architectures, and we apply the minimum-variance estimator to both targeted FEP and our approach. Reference values for the last three systems are obtained using MBAR [Shirts and Chodera, 2008]. Results in Table 1 demonstrate that our approach consistently outperforms Target FEP. Our method's advantage over targeted FEP likely stems from the latter's reliance on instantaneous changes of variables, making it more susceptible to discretization errors. As discussed in Section 3.4, our approach offers inherent robustness to such errors while also eliminating divergence calculations for improved computational efficiency.

**Comparison with Neural TI [Máté et al., 2024a,b].** We report the results of Neural TI on GMM-40 and LJ-79. For the accuracy of Neural TI, it is crucial to ensure the learned energy matches the sample distribution along the entire interpolant path. Therefore, Máté et al. [2024b] proposed to parameterize the energy network using the energy of state $A$ and $B$ as preconditioning. To have a fair comparison, we report Neu-

Table 2: Neural TI with and without preconditioning.

|  | GMM-40 | LJ-79 |
|---|---|---|
| w/ Precond | $0.1 \pm 0.2$ | $356.9 \pm 1.8$ |
| w/o Precond | $-181.6 \pm 6.7$ | $468.8 \pm 391.2$ |

ral TI both with and without preconditioning. Details of the preconditioning design are provided in Appendix F.5. Table 2 shows that Neural TI relies heavily on such problem-specific preconditioning, which constrains flexibility and can be costly when the energy evaluation is expensive.

**Different estimators and training dynamics.** We visualize the estimates using only forward or backward simulation in Equation (27), and the minimum-variance form in Equation (30) throughout training for LJ-128 and ALDP in Figure 2. Our method converges rapidly on both systems, with the minimum-variance estimator clearly outperforming the forward or backward-only estimators.

**Application on umbrella sampling.** A valuable application of our method is umbrella sampling for free energy surface estimation (potential of mean force) in collective variable (CV) space. Traditionally addressed via weighted histogram analysis method [WHAM, Kumar et al., 1992], this

approach requires defining a sequence of "umbrellas" by adding harmonic potentials along the CV dimension to the target potential, then sampling from these umbrellas using MCMC. To correctly aggregate samples from different umbrellas projected onto CV space, we must estimate relative free energies between umbrella potentials for proper reweighting. Our approach integrates naturally into this pipeline by efficiently estimating free energy differences between umbrella pairs.

We demonstrate this with $\varphi^4$ quantum field theory, also studied in Albergo and Vanden-Eijnden [2024] for sampling tasks. The variables are field configurations $\varphi \in \mathbb{R}^{L \times L}$ and we estimate the average magnetization histogram (see Appendix F.1.5 for energy details). The lattice exists in an ordered phase where neighboring sites correlate with the same sign and magnitude, creating the bimodal distribution shown in Figure 3. We estimate the magnetization histogram by performing two umbrella sampling runs—one biased toward negative magnetization and another toward positive magnetization—then combine them by reweighting according to the free energy difference estimated by our method (calculation details in Appendix F.1.5). As illustrated in Figure 4, the reweighted average magnetization distribution successfully recovers the symmetrical nature of the $\varphi^4$ energy.

**One-sided FEAT and large-scale experiments.** We evaluate FEAT on two larger systems—alanine tetrapeptide (ALA-4) and Chignolin—to estimate the solvation free energy. The standard stochastic interpolant struggles to fit larger molecular systems, so we adopt one-sided FEAT to learn the absolute free energies of each system and take their difference, as described in Section 3.6. Leveraging the diffusion-model design of

Table 3: One-sided FEAT on ALA-4 and CHIG.

| | ALA-4 | CHIG |
|---|---|---|
| Reference | 107.5 | 320.19 |
| FEAT | $109.91 \pm 2.55$ | $320.02 \pm 0.70$ |

Karras et al. [2022], our model fits both systems well. As shown in Table 3, the proposed FEAT delivers accurate predictions, demonstrating its scalability and strong potential.

**Runtime discussion**. We report FEAT's inference time in Table 4. Relative to Target FEP (FM), FEAT avoids costly divergence evaluations, significantly improving efficiency.

## 5    Conclusions and Limitations

Free energy difference estimation between states remains a fundamental challenge with wide-ranging applications, yet research in the modern machine learning context has predominantly focused on equilibrium approaches, leaving non-equilibrium methods largely unexplored. Our Free Energy Estimators with Adaptive Transport (FEAT) address this gap by leveraging the stochastic interpolant framework to learn transports that permit both equilibrium and non-equilibrium estimation of the free energy difference through the escorted Jarzynski equality and the Crooks theorem.

One caveat of FEAT is the variance of estimator can be large even with the minimum variance estimator, especially for larger-scale systems. This is because FEAT is based on *importance sampling over the path space*, while target FEP is based on *importance sampling in the state space*. By the data-processing inequality, the overlap between two distributions will not decrease when lifted from state space to path space. Therefore, FEAT and Target FEP with flow matching may be understood as different points on a bias-variance spectrum: FEAT is asymptotically unbiased but tends to exhibit higher variance. Recent work by Schebek et al. [2025] evaluated FEAT and Target FEP for condensed-phase systems, and highlights this point with a detailed discussion.

Our current approach requires access to samples from both states of interest. Future work could explore approaches that relax this requirement, such as those proposed by Vargas et al. [2024], Albergo and Vanden-Eijnden [2024], potentially enabling free energy estimation in settings with limited sampling access. However, these sampling techniques still face notable challenges in scalability, stability, and mode collapsing. How to resolve these issues efficiently remains an open problem. Investigating the generalizability of our approach is another promising direction, potentially with transferable networks as demonstrated by Klein and Noé [2024]. We present a primary demonstration on a toy example in Appendix E.3, and leave molecular systems exploration to future works.

Finally, recent work has investigated the computational complexity of the Jarzynski equality [Guo et al., 2025]. Extending their analysis to our framework would provide additional insight into requirements for reliable estimation.

## Broader Impact

This work aims to estimate the free energy difference, a core quantity in studying chemical reactions, phase transitions, and biomolecular conformational changes. We expect it to have positive impacts in accelerating drug and materials discovery. However, more efficient free energy estimation also carries the risk of enabling the development of harmful chemicals or toxins. We therefore advocate for their responsible use.

## Acknowledgments

We thank Yanze Wang for the discussions on MBAR setup and John D. Chodera for pointing us to relevant history of MBAR. JH acknowledges support from the University of Cambridge Harding Distinguished Postgraduate Scholars Programme. CPG and YD acknowledge the support of Schmidt Sciences programs, an AI2050 Senior Fellowship and Eric and Wendy Schmidt AI in Science Postdoctoral Fellowships; the National Science Foundation (NSF); the National Institute of Food and Agriculture (USDA/NIFA); the Air Force Office of Scientific Research (AFOSR). YW acknowledges support from the Schmidt Science Fellowship, in partnership with the Rhodes Trust, as well as the Simons Center for Computational Physical Chemistry at New York University. YW thanks the high-performance computing resources at Memorial Sloan Kettering Cancer Center and the Washington Square and Abu Dhabi campuses of New York University—we are especially grateful to the technical supporting teams. YW has limited financial interests in Flagship Pioneering, Inc. and its subsidiaries. JMHL acknowledges support from a Turing AI Fellowship under grant EP/V023756/1.

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

# FEAT: Free energy Estimators with Adaptive Transport Appendix

# A  Algorithm

---

**Algorithm 1** Free energy estimation with FEAT

---

**Input:** Energy function of both states $U_a$ and $U_b$ and their samples $x_a^{(1:N)} \sim \mu_a$, $x_b^{(1:N)} \sim \mu_b$; learning rate $\eta_{\text{lr}}$, SDE solver step size $\Delta t$.

**Output:** Free energy difference estimator $\Delta \hat{F} \approx -\log Z_b/Z_a$.

  # initialization:
Randomly initialize vector field $u_t^\psi$ and score network $s_t^\theta$ (or energy network $U_t^\theta$);
  # training:
**repeat** $\theta \leftarrow \theta - \eta_{\text{lr}} \nabla_\theta \left( \mathcal{L}_U^{\text{DSM}} + \mathcal{L}_U^{\text{TSM, 0}} + \mathcal{L}_U^{\text{TSM, 1}} \right)$; $\psi \leftarrow \psi - \eta_{\text{lr}} \nabla_\psi \left( \mathcal{L}_u \right)$; **until** convergence;
  # simulation:
Simulate the forward eq. (34) from $\overrightarrow{X}_0^{(1:N)} = x_a^{(1:N)}$; calculate $\widetilde{W}^v(\overrightarrow{X}^{(1:N)})$ by eq. (33);
Simulate the backward eq. (35) from $\overleftarrow{X}_1^{(1:N)} = x_b^{(1:N)}$; calculate $\widetilde{W}^v(\overleftarrow{X}^{(1:N)})$ by eq. (33);
  # estimation:
initialize $C$ to be the average value of the estimators in eq. (27);
**repeat** calculate $\Delta \hat{F}$ with eq. (30); and set $C \leftarrow \Delta \hat{F}$; **until** convergence.

---

# B  Extended review of free energy estimation methods

Here, we provide an extended review of free energy estimation methods. This section complements the background in Section 2 by offering more details on FEP, target FEP, and BAR, along with a review of MBAR and TI.

**Free energy perturbation** [FEP, Zwanzig, 1954] leverages importance sampling to estimate free energy difference:

$$\Delta F = -- = \log \frac{Z_b}{Z_a} = -\log \frac{1}{Z_a} \int \exp(-U_b(x)) \mathrm{d}x \tag{37}$$

$$= -\log \frac{1}{Z_a} \int \exp(-(U_b(x) - U_a(x) + U_a(x))) \mathrm{d}x \tag{38}$$

$$= -\log \int \exp(-(U_b(x) - U_a(x))) p_a(x) \mathrm{d}x \tag{39}$$

$$= -\log \mathbb{E}_a \left[ \exp(U_a - U_b) \right] \tag{40}$$

where $\mathbb{E}_a$ denotes expectation w.r.t. the equilibrium distribution $\mu_a(\mathrm{d}x) = Z_a^{-1} e^{-U_a(x)} \mathrm{d}x$ of state $S_a$, and $p_a$ is the density of $\mu_a$.

**Bennett acceptance ratio** [BAR, Bennett, 1976] extends the FEP equation into a general form:

$$\Delta F = -\log \frac{Z_b}{Z_a} = -\log \frac{\mathbb{E}_a \left[ \phi(U_b - U_a - C) \right]}{\mathbb{E}_b \left[ \phi(U_a - U_b + C) \right]} + C \tag{41}$$

where it has been proven the minimum variance free energy estimator can be obtained by setting $\phi = 1/(1 + \exp(x))$ and $C = \Delta F$. This estimator was first proposed by Bennett [1976] without detailed proof. Meng and Wong [1996] termed the ratio estimator as "bridge sampling" and proved that the BAR form minimizes MSE using Cauchy–Schwarz inequality. Generalizing this estimator and the proof by Meng and Wong [1996] to non-equilibrium settings, we obtain Proposition 3.2.

**Multi-state Bennett acceptance ratio** [MBAR, Shirts and Chodera, 2008] method $K > 2$ states, which are typically chosen as interpolants between two ends, so that each adjacent pair of distributions have enough overlap. Specifically, we are interested in estimating the free energy difference between all pairs of distributions:

$$\Delta F_{ij} = F_j - F_i = -\log \frac{Z_i}{Z_j} \tag{42}$$

To this end, MBAR bases on the detailed balance between all pairs:

$$Z_i \mathbb{E}_i \big[ \alpha_{ij} \exp(-U_j) \big] = Z_j \mathbb{E}_j \big[ \alpha_{ij} \exp(-U_i) \big] \tag{43}$$

Assume for each distribution $i$, we use $N_i$ samples $\{X_{in}\}_{n=1 \cdots N_i}$. The optimal form of $\alpha_{ij}$ is

$$\alpha_{ij} = \frac{N_j Z_j^{-1}}{\sum_{k=1}^{K} N_k Z_k^{-1} \exp(-U_k)} \tag{44}$$

This is similar to BAR, where the minimal-variance estimator of the free energy differences contains the value to be estimated. Therefore, MBAR also takes an iterative form. Making all the necessary substitutions, we obtain:

$$\hat{F}_i = -\log \sum_{j=1}^{K} \sum_{n=1}^{N_j} \frac{\exp(-U_i(X_{jn}))}{\sum_{k=1}^{K} N_k \exp(\hat{F}_k - U_k X_{jn})} \tag{45}$$

Tan [2004] first generalized the "bridge sampling" framework [Meng and Wong, 1996] to multiple states and derived the optimal form of $\alpha$. Shirts and Chodera [2008] then combined this formulation with the BAR estimator across multiple states, leading to the well-known MBAR method.

**Targeted FEP** [Jarzynski, 2002] method aims to design an invertible transformation $T$ that maps $x_a \sim \mu_a$ to $\mu_{b'}$ such that $\mu_{b'}$ largely overlaps with $\mu_b$. This allows us to use the change of variable formula to track the free energy difference:

$$\mathbb{E}_a \big[ \exp(-\Phi_F) \big] = \exp(-\Delta F), \quad \Phi_F(x) = U_b(T(x)) - U_a(x) - \log |\nabla T(x)| \tag{46}$$

where $|\nabla T(x)|$ is the determinant of the Jacobian matrix of the function $T$. Similarly, as the transformation is invertible, we can write in the reverse direction:

$$\mathbb{E}_b \big[ \exp(-\Phi_R) \big] = \exp(-\Delta F), \quad \Phi_R(x) = U_a(T^{-1}(x)) - U_b(x) - \log |\nabla T^{-1}(x)| \tag{47}$$

**Thermodynamic Integration (TI)** approach also introduces a sequence of distributions that connects the two marginal distributions and estimate free energy difference using the following equality:

$$\Delta F = \int_0^1 \frac{\partial F_t}{\partial t} \mathrm{d}t \tag{48}$$

$$= -\int_0^1 \frac{\frac{\partial Z_t}{\partial t}}{Z_t} \mathrm{d}t \tag{49}$$

$$= -\int_0^1 \frac{\int \exp(-U_t)(-\frac{\partial U_t}{\partial t}) \mathrm{d}x}{\int \exp(-U_t) \mathrm{d}x} \mathrm{d}t \tag{50}$$

$$= \int_0^1 \mathbb{E}_{p_t} \left[ \frac{\partial U_t}{\partial t} \right] \mathrm{d}t \tag{51}$$

## C  Proofs

### C.1  Variational Bounds (Equation (29)) and IWAE Bounds (Equation (28))

We restate the bounds for easier reference:

Let $\widetilde{W}^v$ denote the generalized work associated with samples from the forward and backward SDEs:

$$\mathrm{d}X_t = -\sigma_t^2 \nabla U_t^\theta(X_t)\mathrm{d}t + v_t^\psi(X_t)\mathrm{d}t + \sqrt{2}\sigma_t \overrightarrow{\mathrm{d}B_t}, \quad X_0 \sim \mu_a, \tag{52}$$

$$\mathrm{d}X_t = \sigma_t^2 \nabla U_t^\theta(X_t)\mathrm{d}t + v_t^\psi(X_t)\mathrm{d}t + \sqrt{2}\sigma_t \overleftarrow{\mathrm{d}B_t}, \quad X_1 \sim \mu_b, \tag{53}$$

We then have

$$\mathbb{E}\left[ \log \frac{1}{N} \sum_{n=1}^{N} \exp(\widetilde{W}^v(\overleftarrow{X}^{(n)})) \right] \leq \Delta F \leq -\mathbb{E}\left[ \log \frac{1}{N} \sum_{n=1}^{N} \exp(-\widetilde{W}^v(\overrightarrow{X}^{(n)})) \right] \tag{54}$$

and

$$\mathbb{E}_{\overleftarrow{\mathbb{P}}^v}[\widetilde{W}^v] \leq \Delta F \leq \mathbb{E}_{\overrightarrow{\mathbb{P}}^v}[\widetilde{W}^v] \tag{55}$$

*Proof.* These bounds are corollary based on the escorted Crooks theorem, which says

$$\frac{d\overleftarrow{\mathbb{P}}^v}{d\overrightarrow{\mathbb{P}}^v}(X) = \exp(-\widetilde{W}^v(X) + \Delta F) \Rightarrow \log \frac{d\overleftarrow{\mathbb{P}}^v}{d\overrightarrow{\mathbb{P}}^v}(X) = -\widetilde{W}^v(X) + \Delta F \tag{56}$$

We first look at the ELBO and EUBO bounds, and then we prove the IWAE bounds.

Taking the expectations over escorted Crooks theorem, we obtain

$$\mathbb{E}_{\overrightarrow{\mathbb{P}}^u}\left[\log \frac{d\overleftarrow{\mathbb{P}}^u}{d\overrightarrow{\mathbb{P}}^u}\right] = \mathbb{E}_{\overrightarrow{\mathbb{P}}^u}[-\widetilde{W}^v] + \Delta F, \quad \mathbb{E}_{\overleftarrow{\mathbb{P}}^u}\left[\log \frac{d\overrightarrow{\mathbb{P}}^u}{d\overleftarrow{\mathbb{P}}^u}\right] = \mathbb{E}_{\overleftarrow{\mathbb{P}}^u}[\widetilde{W}^v] - \Delta F \tag{57}$$

We recognize their LHS are negative KL divergences, which are always non-positive. Therefore,

$$\mathbb{E}_{\overrightarrow{\mathbb{P}}^u}[-\widetilde{W}^v] + \Delta F \leq 0, \quad \mathbb{E}_{\overleftarrow{\mathbb{P}}^u}[\widetilde{W}^v] - \Delta F \leq 0 \tag{58}$$

Rearrange these equations, we obtain

$$\mathbb{E}_{\overleftarrow{\mathbb{P}}^u}[\widetilde{W}^v] \leq \Delta F \leq \mathbb{E}_{\overrightarrow{\mathbb{P}}^u}[\widetilde{W}^v], \tag{59}$$

which finishes the proof. $\qquad\square$

The IWAE bound can then be obtained by Jensen's inequality. The green color indicates the terms in Equations (28) and (29) for a clearer visualization.

$$\boxed{\Delta F} = \log \mathbb{E}_{X \sim \overleftarrow{\mathbb{P}}^u}\left[\exp(\widetilde{W}^v(X))\right] \tag{60}$$

$$= \log \mathbb{E}_{X^{(1:N)} \sim \overleftarrow{\mathbb{P}}^u}\left[\frac{1}{N}\sum_{n=1}^N \exp(\widetilde{W}^v(X^{(n)}))\right] \tag{61}$$

$$\geq \boxed{\mathbb{E}_{X^{(1:N)} \sim \overleftarrow{\mathbb{P}}^u}\left[\log \frac{1}{N}\sum_{n=1}^N \exp(\widetilde{W}^v(X^{(n)}))\right]} \tag{62}$$

$$\geq \mathbb{E}_{X^{(1:N)} \sim \overleftarrow{\mathbb{P}}^u}\frac{1}{N}\sum_{n=1}^N \left[\log \exp(\widetilde{W}^v(X^{(n)}))\right] \tag{63}$$

$$= \boxed{\mathbb{E}_{\overleftarrow{\mathbb{P}}^u}[\widetilde{W}^v]} \tag{64}$$

$$\boxed{\Delta F} = -\log \mathbb{E}_{X \sim \overrightarrow{\mathbb{P}}^u}\left[\exp(-\widetilde{W}^v(X))\right] \tag{65}$$

$$= -\log \mathbb{E}_{X^{(1:N)} \sim \overrightarrow{\mathbb{P}}^u}\left[\frac{1}{N}\sum_{n=1}^N \exp(-\widetilde{W}^v(X^{(n)}))\right] \tag{66}$$

$$\leq \boxed{-\mathbb{E}_{X^{(1:N)} \sim \overrightarrow{\mathbb{P}}^u}\left[\log \frac{1}{N}\sum_{n=1}^N \exp(-\widetilde{W}^v(X^{(n)}))\right]} \tag{67}$$

$$\leq -\mathbb{E}_{X^{(1:N)} \sim \overrightarrow{\mathbb{P}}^u}\frac{1}{N}\sum_{n=1}^N \left[\log \exp(-\widetilde{W}^v(X^{(n)}))\right] \tag{68}$$

$$= \boxed{\mathbb{E}_{\overrightarrow{\mathbb{P}}^u}[\widetilde{W}^v]} \tag{69}$$

We hence can see that IWAE bounds are tighter compared to the ELBO and EUBO bounds.

## C.2 Minimum Variance Non-equilibrium Free Energy Estimator (Proposition 3.2)

**Proposition 3.2** (Minimum variance non-equilibrium free energy estimator). *Let $\widetilde{W}_{a\to b}$ and $\widetilde{W}_{b\to a}$ be the work terms defined in Corollary 3.1. The minimum-variance estimator is given by:*

$$\Delta F \approx \log \frac{\sum_{m=1}^{N} \phi(-\widetilde{W}^v(\overleftarrow{X}^{(m)}) + C)}{\sum_{n=1}^{N} \phi(\widetilde{W}^v(\overrightarrow{X}^{(n)}) - C)} + C, \quad \overrightarrow{X}^{(1:N)} \sim \overrightarrow{\mathbb{P}}^v, \quad \overleftarrow{X}^{(1:N)} \sim \overleftarrow{\mathbb{P}}^v \tag{70}$$

*where $\phi$ is the Fermi function $\phi(\cdot) = 1/(1 + \exp(\cdot))$ and $C = \Delta F$ in optimal.*

*Proof.* Our proof follows Bennett [1976], Meng and Wong [1996] closely, with a slight extension to path measures. To increase the readability of the proof, we first summarize the entire structure:

1. express the normalizing factor ratio;

2. express $\text{MSE}^2$ of the normalizing factor ratio estimator, and approximate it with $\delta$-method;

3. apply Cauchy-Schwartz inequality to obtain a lower bound of $\text{MSE}^2$. This gives an optimal condition to minimize $\text{MSE}^2$.

4. plug the condition back to the normalizing factor ratio expression and finish the proof.

---
**1. express the normalizing factor ratio:**

---

First, we have the following equality:

$$Z_a \mathbb{E}_{\mathbb{P}_{a\to b}} \left[ \alpha(X) g\left( \widetilde{W}(X) \right) \right] = Z_b \mathbb{E}_{\mathbb{P}_{b\to a}} \left[ \alpha(X) g\left( -\widetilde{W}(X) \right) \right] \tag{71}$$

where $\alpha$ is an arbitrary function, and $g$ is any function such that $g(r)/g(-r) = \exp(-r)$. Our goal is to find a form of $\alpha$ to minimize the variance (MSE) of the estimator of the ratio between normalization factors. Also, to make things clearer, we explicitly write the direction in the path measure $\mathbb{P}_{a\to b}, \mathbb{P}_{b\to a}$, and note that we drop the superscript ($^v$) for simplicity.

The equality Equation (71) holds, because:

$$Z_a \mathbb{E}_{\mathbb{P}_{a\to b}} \left[ \alpha(X) g\left( \widetilde{W}(X) \right) \right] \tag{72}$$

$$= Z_a \mathbb{E}_{\mathbb{P}_{a\to b}} \left[ \alpha(X) g\left( -\widetilde{W}(X) \right) \exp\left( -\widetilde{W}(X) \right) \right] \quad \triangleright \text{ as } g(r) = g(-r)\exp(-r) \tag{73}$$

$$= Z_a \mathbb{E}_{\mathbb{P}_{a\to b}} \left[ \alpha(X) g\left( -\widetilde{W}(X) \right) \exp\left( \log \frac{d\mathbb{P}_{b\to a}}{d\mathbb{P}_{a\to b}}(X) + \log \frac{Z_b}{Z_a} \right) \right] \tag{74}$$

$$= Z_a \frac{Z_b}{Z_a} \mathbb{E}_{\mathbb{P}_{a\to b}} \left[ \alpha(X) g\left( -\widetilde{W}(X) \right) \frac{d\mathbb{P}_{b\to a}}{d\mathbb{P}_{a\to b}}(X) \right] \tag{75}$$

$$= Z_b \mathbb{E}_{\mathbb{P}_{b\to a}} \left[ \alpha(X) g\left( -\widetilde{W}(X) \right) \right] \tag{76}$$

---
**2. express $\text{MSE}^2$ of the normalizing factor ratio estimator, and approximate it with $\delta$-method:**

---

We now write down its Monte Carlo form. For simplicity, we assume we use the same number of samples from $X_{a\to b}^{(1:N)} \sim \mathbb{P}_{a\to b}$ and $X_{b\to a}^{(1:N)} \sim \mathbb{P}_{b\to a}$.

$$Z_a \frac{1}{N} \sum_n \left[ \alpha(X_{a\to b}^{(n)}) g\left( \widetilde{W}(X_{a\to b}^{(n)}) \right) \right] = Z_b \frac{1}{N} \sum_n \left[ \alpha(X_{b\to a}^{(n)}) g\left( -\widetilde{W}(X_{b\to a}^{(n)}) \right) \right] \tag{77}$$

Therefore,

$$\frac{Z_a}{Z_b} \approx \frac{\frac{1}{N}\sum_n \left[\alpha(X_{b\to a}^{(n)})g\left(-\widetilde{W}(X_{b\to a}^{(n)})\right)\right]}{\frac{1}{N}\sum_n \left[\alpha(X_{a\to b}^{(n)})g\left(\widetilde{W}(X_{a\to b}^{(n)})\right)\right]} = \hat{r} \tag{78}$$

We consider the MSE of $\hat{r}$:

$$\mathrm{MSE} = \mathbb{E}[\hat{r} - r]^2/r^2 \tag{79}$$

To simply the calculation of MSE, ket $\bar{\eta}_1$, $\bar{\eta}_2$ be respectively the numerator and denominator of $\hat{r}$:

$$\bar{\eta}_1 = \frac{1}{N}\sum_n \left[\alpha(X_{b\to a}^{(n)})g\left(-\widetilde{W}(X_{b\to a}^{(n)})\right)\right] \tag{80}$$

$$\bar{\eta}_2 = \frac{1}{N}\sum_n \left[\alpha(X_{a\to b}^{(n)})g\left(\widetilde{W}(X_{a\to b}^{(n)})\right)\right] \tag{81}$$

and we denote $\eta_1$ and $\eta_2$ as the true value:

$$\eta_1 = \mathbb{E}_{b\to a}\left[\alpha(X_{b\to a})g\left(-\widetilde{W}(X_{b\to a})\right)\right] \tag{82}$$

$$\eta_2 = \mathbb{E}_{a\to b}\left[\alpha(X_{a\to b})g\left(\widetilde{W}(X_{a\to b})\right)\right] \tag{83}$$

where we write $\mathbb{E}_{b\to a}$ as a shorthand of $\mathbb{E}_{\mathbb{P}_{b\to a}}$. Note that

$$Z_b\eta_1 = Z_a\eta_2 \tag{84}$$

Following Meng and Wong [1996], we use $\delta$-method to approximate the MSE:

$$\mathrm{MSE} = \mathbb{E}[\hat{r} - r]^2/r^2 \approx \frac{V(\bar{\eta}_1)}{\eta_1^2} + \frac{V(\bar{\eta}_2)}{\eta_2^2} \tag{85}$$

The variances are given by:

$$V(\bar{\eta}_1) = \frac{1}{N}\left(\mathbb{E}_{b\to a}\left[\alpha(X_{b\to a})^2 g\left(-\widetilde{W}(X_{b\to a})\right)^2\right] - \mathbb{E}_{b\to a}^2\left[\alpha(X_{b\to a})g\left(-\widetilde{W}(X_{b\to a})\right)\right]\right) \tag{86}$$

$$V(\bar{\eta}_2) = \frac{1}{N}\left(\mathbb{E}_{a\to b}\left[\alpha(X_{a\to b})^2 g\left(\widetilde{W}(X_{a\to b})\right)^2\right] - \mathbb{E}_{a\to b}^2\left[\alpha(X_{a\to b})g\left(\widetilde{W}(X_{a\to b})\right)\right]\right) \tag{87}$$

and hence we have

$$\mathrm{MSE}^2 \approx \frac{\mathbb{E}_{b\to a}\left[\alpha(X_{b\to a})^2 g\left(-\widetilde{W}(X_{b\to a})\right)^2\right] + \frac{Z_a^2}{Z_b^2}\mathbb{E}_{a\to b}\left[\alpha(X_{a\to b})^2 g\left(\widetilde{W}(X_{a\to b})\right)^2\right]}{\mathbb{E}_{b\to a}^2\left[\alpha(X_{b\to a})g\left(-\widetilde{W}(X_{b\to a})\right)\right]} - \frac{2}{N} \tag{88}$$

We now look at the second term in the numerator:

$$\frac{Z_a^2}{Z_b^2}\mathbb{E}_{a\to b}\left[\alpha(X_{a\to b})^2 g\left(\widetilde{W}(X_{a\to b})\right)^2\right] \tag{89}$$

$$= \frac{Z_a^2}{Z_b^2}\mathbb{E}_{a\to b}\left[\alpha(X_{a\to b})^2 g\left(\widetilde{W}(X_{a\to b})\right) g\left(\widetilde{W}(X_{a\to b})\right)\right] \tag{90}$$

$$= \frac{Z_a^2}{Z_b^2}\mathbb{E}_{a\to b}\left[\alpha(X_{a\to b})^2 g\left(\widetilde{W}(X_{a\to b})\right) g\left(-\widetilde{W}(X_{a\to b})\right)\exp\left(-\widetilde{W}(X_{a\to b})\right)\right] \tag{91}$$

$$= \frac{Z_a^2}{Z_b^2}\mathbb{E}_{a\to b}\left[\alpha(X_{a\to b})^2 g\left(\widetilde{W}(X_{a\to b})\right) g\left(-\widetilde{W}(X_{a\to b})\right)\exp\left(\log\frac{d\mathbb{P}_{b\to a}}{d\mathbb{P}_{a\to b}}(X_{a\to b}) + \log\frac{Z_b}{Z_a}\right)\right] \tag{92}$$

$$= \frac{Z_a}{Z_b}\mathbb{E}_{a\to b}\left[\alpha(X_{a\to b})^2 g\left(\widetilde{W}(X_{a\to b})\right) g\left(-\widetilde{W}(X_{a\to b})\right)\frac{d\mathbb{P}_{b\to a}}{d\mathbb{P}_{a\to b}}(X_{a\to b})\right] \tag{93}$$

$$= \frac{Z_a}{Z_b}\mathbb{E}_{b\to a}\left[\alpha(X_{b\to a})^2 g\left(\widetilde{W}(X_{b\to a})\right) g\left(-\widetilde{W}(X_{b\to a})\right)\right] \tag{94}$$

Therefore, we can write the MSE as

$$
\begin{aligned}
&\text{MSE}^2 \\
&\approx \frac{\mathbb{E}_{b \to a}\left[\alpha(X_{b \to a})^2 g\left(-\widetilde{W}(X_{b \to a})\right)\left(g\left(-\widetilde{W}(X_{b \to a})\right) + \frac{Z_a}{Z_b}g\left(\widetilde{W}(X_{b \to a})\right)\right)\right]}{\mathbb{E}_{b \to a}^2\left[\alpha(X_{b \to a})g\left(-\widetilde{W}(X_{b \to a})\right)\right]} - \frac{2}{N}
\end{aligned} \quad (95)
$$

---

**3. apply Cauchy-Schwartz inequality to obtain a lower bound of MSE$^2$:**

---

We denote

$$
A(X) = \alpha(X)^2 g\left(-\widetilde{W}(X)\right)\left(g\left(-\widetilde{W}(X)\right) + \frac{Z_a}{Z_b}g\left(\widetilde{W}(X)\right)\right) \quad (96)
$$

$$
B(X) = g\left(-\widetilde{W}(X)\right) \Big/ \left(g\left(-\widetilde{W}(X)\right) + \frac{Z_a}{Z_b}g\left(\widetilde{W}(X)\right)\right) \quad (97)
$$

We can write the denominator as

$$
\mathbb{E}_{b \to a}^2\left[\alpha(X_{b \to a})g\left(-\widetilde{W}(X_{b \to a})\right)\right] = \mathbb{E}_{b \to a}^2\left[\sqrt{A}\sqrt{B}\right] \quad (98)
$$

and the entire MSE as

$$
\text{MSE} = \mathbb{E}_{b \to a}[A] \Big/ \mathbb{E}_{b \to a}^2\left[\sqrt{A}\sqrt{B}\right] - 2/N \quad (99)
$$

Following Meng and Wong [1996], we apply the Cauchy-Schwartz inequality:

$$
\mathbb{E}_{b \to a}^2\left[\sqrt{A}\sqrt{B}\right] \le \mathbb{E}_{b \to a}[A]\,\mathbb{E}_{b \to a}[B] \quad (100)
$$

Due to the property of $g$, we can see $B$ is always positive:

$$
B(X) = g\left(-\widetilde{W}(X)\right) \Big/ \left(g\left(-\widetilde{W}(X)\right) + \frac{Z_a}{Z_b}g\left(\widetilde{W}(X)\right)\right) \quad (101)
$$

$$
= 1/(1 + Z_a/Z_b \exp(-\widetilde{W}(X))) \quad (102)
$$

We hence have

$$
\mathbb{E}_{b \to a}[A] \Big/ \mathbb{E}_{b \to a}^2\left[\sqrt{A}\sqrt{B}\right] \ge 1/\mathbb{E}_{b \to a}[B] \quad (103)
$$

Note that $\mathbb{E}_{b \to a}[B]$ does not depend on $\alpha$. Hence, we found a lower bound of the MSE w.r.t $\alpha$. The equality holds when $A \propto B$, i.e.,

$$
\alpha(X) \propto \frac{1}{g\left(-\widetilde{W}(X)\right) + Z_a/Z_b\,g\left(\widetilde{W}(X)\right)} \quad (104)
$$

---

**4. plug the condition back to the normalizing factor ratio expression and finish the proof:**

---

Plugging $\alpha$ back to Equation (78), we obtain

$$
\frac{Z_a}{Z_b} \approx \frac{\sum_n\left[\frac{g\left(-\widetilde{W}(X_{b \to a}^{(n)})\right)}{g\left(-\widetilde{W}(X_{b \to a}^{(n)})\right) + Z_a/Z_b\,g\left(\widetilde{W}(X_{b \to a}^{(n)})\right)}\right]}{\sum_n\left[\frac{g\left(\widetilde{W}(X_{a \to b}^{(n)})\right)}{g\left(-\widetilde{W}(X_{a \to b}^{(n)})\right) + Z_a/Z_b\,g\left(\widetilde{W}(X_{a \to b}^{(n)})\right)}\right]} \quad (105)
$$

We now use the property of $g$, namely $g(r)/g(-r) = \exp(-r)$, to simplify both the numerator and the denominator:

$$\frac{g\left(-\widetilde{W}(X_{b\to a}^{(n)})\right)}{g\left(-\widetilde{W}(X_{b\to a}^{(n)})\right) + Z_a/Z_b g\left(\widetilde{W}(X_{b\to a}^{(n)})\right)} \tag{106}$$

$$= \frac{1}{1 + Z_a/Z_b \exp\left(-\widetilde{W}(X_{b\to a}^{(n)})\right)} \tag{107}$$

$$= \frac{1}{1 + \exp(\Delta F) \exp\left(-\widetilde{W}(X_{b\to a}^{(n)})\right)} \tag{108}$$

$$= \frac{1}{1 + \exp\left(-\widetilde{W}(X_{b\to a}^{(n)}) + \Delta F\right)} \tag{109}$$

and

$$\frac{g\left(\widetilde{W}(X_{a\to b}^{(n)})\right)}{g\left(-\widetilde{W}(X_{a\to b}^{(n)})\right) + Z_a/Z_b g\left(\widetilde{W}(X_{a\to b}^{(n)})\right)} \tag{110}$$

$$= \frac{1}{\exp\left(\widetilde{W}(X_{a\to b}^{(n)})\right) + Z_a/Z_b} \tag{111}$$

$$= \frac{\exp(-\Delta F)}{\exp\left(\widetilde{W}(X_{a\to b}^{(n)}) - \Delta F\right) + 1} \tag{112}$$

Taking log on both sides of Equation (105) and plugging in the simplified numerator and denominator, we obtain

$$\Delta F = \log \frac{Z_a}{Z_b} \approx \log \frac{\sum_n \left[\frac{1}{1 + \exp\left(-\widetilde{W}(X_{b\to a}^{(n)}) + \Delta F\right)}\right]}{\sum_n \left[\frac{\exp(-\Delta F)}{\exp\left(\widetilde{W}(X_{a\to b}^{(n)}) - \Delta F\right) + 1}\right]} \tag{113}$$

$$= \log \frac{\sum_n \left[\frac{1}{1 + \exp\left(-\widetilde{W}(X_{b\to a}^{(n)}) + \Delta F\right)}\right]}{\sum_n \left[\frac{1}{\exp\left(\widetilde{W}(X_{a\to b}^{(n)}) - \Delta F\right) + 1}\right]} + \Delta F \tag{114}$$

Let $\phi(\cdot) = 1/(1 + \exp(\cdot))$ as the Fermi function, we have

$$\Delta F \approx \log \frac{\sum_n \phi(-\widetilde{W}(X_{b\to a}^{(n)}) + \Delta F)}{\sum_n \phi(\widetilde{W}(X_{a\to b}^{(n)}) - \Delta F)} + \Delta F \tag{115}$$

which finishes the proof. $\qquad\square$

## C.3 Escorted Jarzynski and Controlled Crooks with Imperfect Boundary Conditions (Corollary 3.1 and Proposition 3.3)

**Corollary 3.1** (Escorted Jarzynski with imperfect boundary conditions). *Given $v_t^\psi$ and $U_t^\theta$, consider the forward and backward SDEs:*

$$\mathrm{d}X_t = -\sigma_t^2 \nabla U_t^\theta(X_t)\mathrm{d}t + v_t^\psi(X_t)\mathrm{d}t + \sqrt{2}\sigma_t \overrightarrow{\mathrm{d}B_t}, \quad X_0 \sim \mu_a, \tag{116}$$

$$\mathrm{d}X_t = \sigma_t^2 \nabla U_t^\theta(X_t)\mathrm{d}t + v_t^\psi(X_t)\mathrm{d}t + \sqrt{2}\sigma_t \overleftarrow{\mathrm{d}B_t}, \quad X_1 \sim \mu_b, \tag{117}$$

where $\sigma_t \geq 0$ and $\mu_a$ and $\mu_b$ denote the distributions associated with the energies $U_a$ and $U_b$, respectively. Define also the "corrected generalized work":

$$\widetilde{W}^v(X) = \int_0^1 \left( -\nabla \cdot v_t^\psi(X_t) + \nabla U_t^\theta(X_t) \cdot v_t^\psi(X_t) + \partial_t U_t^\theta(X_t) \right) \mathrm{d}t$$
$$+ \log \underbrace{\frac{\exp(-U_a(X_0))\exp(-U_1^\theta(X_1))}{\exp(-U_b(X_1))\exp(-U_0^\theta(X_0))}}_{\text{correction term}} \tag{118}$$

*Using the generalized work with correction, we have the same escorted Jarzynski equality as before:*

$$\Delta F = -\log \mathbb{E}_{\overrightarrow{\mathbb{P}}^v}[\exp(-\widetilde{W}^v)] = \log \mathbb{E}_{\overleftarrow{\mathbb{P}}^v}[\exp(\widetilde{W}^v)] \tag{119}$$

*where $\overrightarrow{\mathbb{P}}^v$ and $\overleftarrow{\mathbb{P}}^v$ are the path measures over the solutions to the forward and backward SDE, respectively.*

**Proposition 3.3** (Discretized Controlled Crooks theorem with imperfect boundary conditions)**.** *Let $\mathcal{N}^+$ and $\mathcal{N}^-$ be as in Equations* (31) *and* (32)*, and define the forward and backward discretized paths via*

$$\overrightarrow{X}_{t_{i+1}} = \overrightarrow{X}_{t_i} - \sigma_{t_i}^2 \nabla U_{t_i}^\theta(\overrightarrow{X}_{t_i})\Delta t_i + v_{t_i}^\psi(\overrightarrow{X}_{t_i})\Delta t_i + \sqrt{2\Delta t_i}\sigma_{t_i}\eta_i, \qquad \overrightarrow{X}_0 \sim \mu_a, \quad (120)$$

$$\overleftarrow{X}_{t_{i-1}} = \overleftarrow{X}_{t_i} - \sigma_{t_i}^2 \nabla U_{t_i}^\theta(\overleftarrow{X}_{t_i})\Delta t_{i-1} - v_{t_i}^\psi(\overleftarrow{X}_{t_i})\Delta t_{i-1} + \sqrt{2\Delta t_{i-1}}\sigma_{t_i}\eta_i, \qquad \overleftarrow{X}_1 \sim \mu_b, \quad (121)$$

*where we assume that $\sigma_t > 0$, $\Delta t_i = t_{i+1} - t_i$ and $\eta_i \sim N(0, \mathit{Id})$, independent for each $i = 0, \dots, M$. Then, we have:*

$$\Delta F = -\log \mathbb{E}\left[ \frac{\exp(-U_b(\overrightarrow{X}_1)) \prod_{i=1}^M \mathcal{N}^-(\overrightarrow{X}_{t_{i-1}} | \overrightarrow{X}_{t_i})}{\exp(-U_a(\overrightarrow{X}_0)) \prod_{i=1}^M \mathcal{N}^+(\overrightarrow{X}_{t_i} | \overrightarrow{X}_{t_{i-1}})} \right] \tag{122}$$

$$= \log \mathbb{E}\left[ \frac{\exp(-U_a(\overleftarrow{X}_0)) \prod_{i=1}^M \mathcal{N}^+(\overleftarrow{X}_{t_i} | \overleftarrow{X}_{t_{i-1}})}{\exp(-U_b(\overleftarrow{X}_1)) \prod_{i=1}^M \mathcal{N}^-(\overleftarrow{X}_{t_{i-1}} | \overleftarrow{X}_{t_i})} \right] \tag{123}$$

We first prove Corollary 3.1.

*Proof.* First, we note that the escorted Jarzynski $\Delta F = -\log \mathbb{E}_{\overrightarrow{\mathbb{P}}^v}[\exp(-\widetilde{W}^v)] = \log \mathbb{E}_{\overleftarrow{\mathbb{P}}^v}[\exp(\widetilde{W}^v)]$ can be obtained from controlled Crooks theorem. Therefore, to show Equation (119), we only need to prove:

$$\int_0^1 \left( -\nabla \cdot v_t^\psi(X_t) + \nabla U_t^\theta(X_t) \cdot v_t^\psi(X_t) + \partial_t U_t^\theta(X_t) \right) \mathrm{d}t$$
$$+ \log \frac{\exp(-U_a(X_0))\exp(-U_1^\theta(X_1))}{\exp(-U_b(X_1))\exp(-U_0^\theta(X_0))} = \Delta F - \log \frac{\mathrm{d}\overleftarrow{\mathbb{P}}^v}{\mathrm{d}\overrightarrow{\mathbb{P}}^v}(X) \quad (124)$$

Consider the SDEs as Equations (23) and (24). However, instead of starting from $U_a$ and $U_b$, we start from $U_0^\theta$ and $U_1^\theta$. We define their path measures as $\overrightarrow{\mathbb{P}}^{v\prime}$ and $\overleftarrow{\mathbb{P}}^{v\prime}$, and we have

$$\frac{\mathrm{d}\overleftarrow{\mathbb{P}}^{v\prime}}{\mathrm{d}\overrightarrow{\mathbb{P}}^{v\prime}}(X) = \frac{\mathrm{d}\overleftarrow{\mathbb{P}}^v}{\mathrm{d}\overrightarrow{\mathbb{P}}^v}(X) \cdot \frac{\exp(-U_a(X_0))/Z_a}{\exp(-U_b(X_1))/Z_b} \cdot \frac{\exp(-U_1^\theta(X_1))/Z_{U_1^\theta}}{\exp(-U_0^\theta(X_0))/Z_{U_0^\theta}} \tag{125}$$

According to the controlled Crooks theorem (as in Equation (16)) applied to the transport between $U_1^\theta$ and $U_0^\theta$, we have

$$\int_0^1 \left( -\nabla \cdot v_t^\psi + \nabla U_t^\theta \cdot v_t^\psi + \partial_t U_t^\theta \right) \mathrm{d}t = -\log \frac{Z_{U_1^\theta}}{Z_{U_0^\theta}} - \log \frac{\mathrm{d}\overleftarrow{\mathbb{P}}^{v\prime}}{\mathrm{d}\overrightarrow{\mathbb{P}}^{v\prime}}(X) \tag{126}$$

Take logarithm of Equation (125) and add with Equation (126), we obtain

$$\int_0^1 \left( -\nabla \cdot v_t^\psi + \nabla U_t^\theta \cdot v_t^\psi + \partial_t U_t^\theta \right) \mathrm{d}t + \log \frac{\exp(-U_a(X_0))\exp(-U_1^\theta(X_1))}{\exp(-U_b(X_1))\exp(-U_0^\theta(X_0))}$$
$$- \underbrace{\log \frac{Z_a}{Z_b}}_{\Delta F} - \log \frac{\cancel{Z_{U_1^\theta}}}{\cancel{Z_{U_0^\theta}}} = -\log \frac{\cancel{Z_{U_1^\theta}}}{\cancel{Z_{U_0^\theta}}} - \log \frac{\mathrm{d}\overleftarrow{\mathbb{P}}^v}{\mathrm{d}\overrightarrow{\mathbb{P}}^v}(X) \quad (127)$$

which finishes the proof of Corollary 3.1. $\qquad\square$

Next, we look at Appendix C.3.

*Proof.* We will only provide proof for Equation (122) in the following, and Equation (123) follows exactly the same principle.

As the Gaussian kernel in Equations (120) and (121) is normalized, we have

$$\int \exp(-U_b(\vec{X}_1)) \prod_{i=1}^{M} \mathcal{N}^-(\vec{X}_{t_{i-1}}|\vec{X}_{t_i}) \mathrm{d}\vec{X} = Z_b, \tag{128}$$

$$\int \exp(-U_a(\vec{X}_0)) \prod_{i=1}^{M} \mathcal{N}^+(\vec{X}_{t_i}|\vec{X}_{t_{i-1}}) \mathrm{d}\vec{X} = Z_a \tag{129}$$

Therefore,

$$-\log \mathbb{E}\left[\frac{\exp(-U_b(\vec{X}_1)) \prod_{i=1}^{M} \mathcal{N}^-(\vec{X}_{t_{i-1}}|\vec{X}_{t_i})}{\exp(-U_a(\vec{X}_0)) \prod_{i=1}^{M} \mathcal{N}^+(\vec{X}_{t_i}|\vec{X}_{t_{i-1}})}\right] \tag{130}$$

$$= -\log \int \frac{\exp(-U_a(\vec{X}_0)) \prod_{i=1}^{M} \mathcal{N}^+(\vec{X}_{t_i}|\vec{X}_{t_{i-1}})}{Z_a} \frac{\exp(-U_b(\vec{X}_1)) \prod_{i=1}^{M} \mathcal{N}^-(\vec{X}_{t_{i-1}}|\vec{X}_{t_i})}{\exp(-U_a(\vec{X}_0)) \prod_{i=1}^{M} \mathcal{N}^+(\vec{X}_{t_i}|\vec{X}_{t_{i-1}})} \mathrm{d}\vec{X} \tag{131}$$

$$= -\log \frac{\int \exp(-U_b(\vec{X}_1)) \prod_{i=1}^{M} \mathcal{N}^-(\vec{X}_{t_{i-1}}|\vec{X}_{t_i}) \mathrm{d}\vec{X}}{Z_a} \tag{132}$$

$$= -\log \frac{Z_b}{Z_a} = \Delta F \tag{133}$$

$\square$

## C.4 Escorted Jarzynski Equality with ODE Transport

We restate the equivalence of escorted Jarzynski equality with ODE transport and target FEP formula with the instantaneous change of variables in the following:

Let $X_t$ evolve according to the ODE

$$\mathrm{d}X_t = v_t(X_t)\,\mathrm{d}t, \quad X_0 \sim \mu_a. \tag{134}$$

For a smooth energy function $U_t$ with, potentially $U_0 \neq U_a$ and $U_1 \neq U_b$, the escorted Jarzynski equation with imperfect boundary conditions (Corollary 3.1) holds:

$$\Delta F = -\log \mathbb{E}_{X_0 \sim \mu_a}\left[\exp(-\widetilde{W}^v)\right],$$

$$\widetilde{W}^v = \int_0^1 (-\nabla \cdot v_t + \nabla U_t \cdot v_t + \partial_t U_t)\,\mathrm{d}t + \log \frac{\exp(-U_a(X_0))\exp(-U_1(X_1))}{\exp(-U_b(X_1))\exp(-U_0(X_0))} \tag{135}$$

This is equivalent to the target FEP formula with the instantaneous change of variables:

$$\Delta F = -\log \mathbb{E}_{X_0 \sim \mu_a}\left[\exp(-\Phi)\right], \quad \Phi(X) = U_b(X_1) - U_a(X_0) - \int_0^1 \nabla \cdot v_t\,\mathrm{d}t. \tag{136}$$

We note that the proof for escorted Jarzynski by Vargas et al. [2024], Albergo and Vanden-Eijnden [2024] requires $\sigma_t \neq 0$. Concretely, the proof by Vargas et al. [2024] relies on the FB RND, which applies only to SDEs. On the other hand, while Proposition 3 by Albergo and Vanden-Eijnden [2024] states that $\sigma_t \geq 0$, the proof essentially requires $\sigma_t \neq 0$ in order to eliminate $\sigma_t$ from both sides in Equation (63) on page 19. One valid proof for $\sigma_t = 0$ was provided by Tian et al. [2024] using the generalized Liouville equation. Here, we consider a more straightforward derivation, which also directly showcases the equivalence to target FEP formula with the instantaneous change of variables.

*Proof.* To prove Equation (135), we directly show the equivalence between $\Phi$ and $\widetilde{W}^v$. To do so, we consider the total derivative of $U_t(X_t)$:

$$\frac{\mathrm{d}U_t(X_t)}{\mathrm{d}t} = \frac{\partial U_t(X_t)}{\partial t} + \nabla U_t(X_t) \cdot \frac{\mathrm{d}X_t}{\mathrm{d}t} = \frac{\partial U_t(X_t)}{\partial t} + \nabla U_t(X_t) \cdot v_t(X_t) \tag{137}$$

The second equality is due to the ODE Equation (134). We therefore have

$$\widetilde{W}^v = \int_0^1 \left(-\nabla \cdot v_t + \nabla U_t \cdot v_t + \partial_t U_t\right) \mathrm{d}t + \log \frac{\exp(-U_a(X_0))\exp(-U_1(X_1))}{\exp(-U_b(X_1))\exp(-U_0(X_0))} \tag{138}$$

$$= \int_0^1 \frac{\mathrm{d}U_t(X_t)}{\mathrm{d}t} - \int_0^1 \nabla \cdot v_t \, \mathrm{d}t + \log \frac{\exp(-U_a(X_0))\exp(-U_1(X_1))}{\exp(-U_b(X_1))\exp(-U_0(X_0))} \tag{139}$$

$$= U_1(X_1) - U_0(X_0) - \int_0^1 \nabla \cdot v_t \, \mathrm{d}t + \log \frac{\exp(-U_a(X_0))\exp(-U_1(X_1))}{\exp(-U_b(X_1))\exp(-U_0(X_0))} \tag{140}$$

$$= U_b(X_1) - U_a(X_0) - \int_0^1 \nabla \cdot v_t \, \mathrm{d}t \tag{141}$$

$$= \Phi(X) \tag{142}$$

which finishes the proof. $\qquad\square$

## C.5   Equivalence between TI and Our Approach with Perfect Transport

When our method is optimally trained—such that the density of samples simulated from the SDE matches the energy $U_t$ at every time step—it becomes equivalent to TI. We only prove this equivalence using forward work, while the backward work will follow the same argument. To show this, let $p_t$ denote the sample density at time step $t$. At optimality, following Albergo and Vanden-Eijnden [2024], we have:

$$\nabla \cdot (v_t p_t) = \partial_t p_t \tag{143}$$
$$\nabla \cdot v_t p_t + v_t \nabla \cdot p_t = \partial_t p_t \tag{144}$$
$$\nabla \cdot v_t p_t - v_t p_t \nabla \cdot U_t = p_t \partial_t \log p_t \tag{145}$$
$$\nabla \cdot v_t - v_t \nabla \cdot U_t = \partial_t \log p_t \tag{146}$$

Then, the generalized work

$$\widetilde{W}^v = \int_0^1 \left(-\nabla \cdot v_t + \nabla U_t \cdot v_t + \partial_t U_t\right) \mathrm{d}t$$

$$= \int_0^1 \left(-\partial_t \log p_t + \partial_t U_t\right) \mathrm{d}t = \int_0^1 -\partial_t Z_t \mathrm{d}t = Z_0 - Z_1 \tag{147}$$

i.e., $\widetilde{W}^v$ will always be a constant under perfect transport. Therefore, we can write the escorted Jarzynski as

$$\Delta F = -\log \mathbb{E}_{\overrightarrow{\mathbb{P}}^v}[\exp(-\widetilde{W}^v)] \tag{148}$$

$$= \mathbb{E}_{\overrightarrow{\mathbb{P}}^v}[\widetilde{W}^v] \qquad \triangleright \text{ as } \widetilde{W}^v \text{ is a constant} \tag{149}$$

$$= \int_0^1 \mathbb{E}_{p_t}[-\partial_t \log p_t + \partial_t U_t] \mathrm{d}t \tag{150}$$

$$= \int_0^1 \underbrace{\mathbb{E}_{p_t}[-\partial_t \log p_t]}_{=\int -\partial_t p_t(x)\mathrm{d}x=0} \mathrm{d}t + \underbrace{\int_0^1 \mathbb{E}_{p_t}[\partial_t U_t] \mathrm{d}t}_{\text{TI formula}} \tag{151}$$

# D   SE(3)-equivariant and -invariant Graph Neural Networks

For $n$-particle problem and molecular systems, we adopt E(n)-equivariant graph neural networks (EGNN) proposed in [Satorras et al., 2021]. Given a 3D graph $G = (V, E, X, H)$ where $V$ is a set of

vertices, $E \subseteq V \times V$ is a set of edges, $X \in \mathbb{R}^{N \times 3}$ is a set of atomic coordinates and $H \in \mathbb{R}^{N \times K}$ is a set of node features with feature dimension $K$. The procedure of EGNN is the following:

$$r_{ij} = x_i^l - x_j^l \tag{152}$$

$$m_i^l = \sum_{j \in \mathcal{N}(i)} \phi_e(h_i^l, h_j^l, \|r_{ij}\|, e_{ij}) \tag{153}$$

$$h_i^{l+1} = \phi_h(h_i^l, m_i^l) \tag{154}$$

$$x_i^{l+1} = x_i^l + \sum_{j \in \mathcal{N}(i)} r_{ij} \phi_x(m_i^l) \tag{155}$$

where we discard the coordinate update when we only need SE(3) invariance. To break the reflection symmetry, we introduce a cross-product during the position update which is reflection-antisymmetric [Du et al., 2022].

$$c_{ij} = \frac{x_i^l \times x_j^l}{\|x_i^l \times x_j^l\|} \tag{156}$$

$$x_i^{l+1} = x_i^l + \sum_{j \in \mathcal{N}(i)} r_{ij} \phi_x(m_i^l) + c_{ij} \phi_c(m_i^l) \tag{157}$$

where $\phi_e$, $\phi_h$, $\phi_x$ and $\phi_c$ are different neural networks to encode edge, node, relative direction and cross direction scalar features.

# E  Additional Experimental Results

## E.1  Runtime Analysis

Table 4: Inference time for FEAT and Target FEP (FM).

| Name | GMM (d=100) | LJ-55 | ALDP-T |
|---|---|---|---|
| FEAT (ours) | 8 s | 2 min | 20 s |
| Target FEP (FM) | 40 s | >10 h | 40 min |

Besides the results in Table 4, we also include a brief discussion against neural TI and MBAR below:

- Versus neural TI, FEAT performs transport from both sides, which can roughly double its runtime. However, we observed that Neural TI requires a much larger sample size, and preconditioning for the network to achieve ideal performance, which largely increase its inference time.

- Versus MBAR, FEAT is a neural network approach that requires training. But it does not need intermediate samples. We here take an example using ALDP. This can make it more favorable when sampling from the intermediate is expensive. For ALDP, generating samples for each target requires about 1 day on our machine. Collecting all the targets for MBAR can hence take between 1-10 days, depending on whether the simulations are run in parallel. In contrast, our training process takes only 1-2 hours, which is significantly faster than sampling from all intermediate densities. However, this comparison can vary depending how the sample collection pipeline are implemented.

## E.2  Robustness of FEAT

In this section, we analyze the robustness of FEAT and Target FEP with flow matching against different sample size and number of discretization steps on GMM with different dimensionalities. We can see FEAT achieves greater robustness toward less steps and less samples. This results also reflect our discussion in Proposition 3.3 for discretization errors.

| Table 5: FEAT | | | |
|---|---|---|---|
| #step | sample size | GMM-40D | GMM-100D |
| 50 | 5000 | -0.05 ± 0.33 | -4.04 ± 1.92 |
| 100 | 5000 | 0.12 ± 0.07 | -6.43 ± 1.72 |
| 500 | 5000 | 0.00 ± 0.06 | -3.59 ± 1.06 |
| 500 | 500 | 0.13 ± 0.09 | -5.56 ± 1.87 |
| 500 | 1000 | -0.04 ± 0.05 | -6.46 ± 1.86 |
| 500 | 5000 | 0.00 ± 0.06 | -3.59 ± 1.06 |

| Table 6: Target FEP | | | |
|---|---|---|---|
| #step | sample size | GMM-40D | GMM-100D |
| 50 | 5000 | -0.98 ± 0.23 | -15.44 ± 6.02 |
| 100 | 5000 | -0.48 ± 0.17 | -12.10 ± 4.80 |
| 500 | 5000 | -0.12 ± 0.08 | -13.72 ± 2.46 |
| 500 | 500 | -0.16 ± 0.37 | -22.24 ± 3.43 |
| 500 | 1000 | 0.02 ± 0.18 | -19.73 ± 2.13 |
| 500 | 5000 | -0.12 ± 0.08 | -13.72 ± 2.46 |

### E.3 Demonstration of Transferable FEAT

FEAT can be trained on several datasets from multiple targets, with conditions on the target parameters. Once trained, this model will allow us to apply FEAT to similar systems without re-training, similar to what has been demonstrated in transferable Boltzmann generators [Klein and Noé, 2024].

We showcase this potential on GMM-40 with different scaling factor. Concretely, we scale the state $B$ with a scalar $0.5, 0.7, 0.9, 1.1, 1.3, 1.5$ when training. The network also takes the scalar as an input. After training, we evaluate on unseen scalars and report the average error and standard deviation across 3 runs. As we can see, the transferable FEAT model achieves good accuracy across a range of unseen targets. Also, as expected, interpolation yields better performance than extrapolation.

Table 7: Transferable FEAT on GMM with different scalars unseen during training.

| Scalar | 0.45 | 0.6 | 0.8 | 1.0 | 1.2 | 1.4 | 1.55 |
|---|---|---|---|---|---|---|---|
| Error ± std | -3.52 ± 1.13 | -0.07 ± 0.05 | -0.02 ± 0.06 | -0.01 ± 0.06 | 0.01 ± 0.04 | 0.003 ± 0.04 | 0.03 ± 0.13 |

## F   Additional Experimental Details

### F.1   Systems

#### F.1.1   Gaussian Mixtures

We consider estimating the free energy differences between two Gaussian Mixtures in 40/100-dimensional space. Our implementation is based on the code by Midgley et al. [2023]. Below are parameters for $S_a$ and $S_b$:

- $S_a$: 16 mixture components, components mean $\sim \mathcal{U}(-2, 2)$, std `softplus`$(-3)$, random seed 10.
- $S_b$: 40 mixture components, components mean $\sim \mathcal{U}(-2, 2)$, std `softplus`$(-2)$, random seed 0.

#### F.1.2   Lennard-Jones (LJ) particles

We consider alchemical free energy for $N = 55/79/128$ LJ particles. We obtain samples from states $S_a$ and $S_b$ with the Metropolis-adjusted Langevin algorithm (MALA) for 100,000 steps. We remove the first 20,000 samples as the burn-in period. The step size is dynamically adjusted on the fly to ensure the acceptance rate is roughly 0.6. Below are detailed settings for $S_a$ and $S_b$:

- $S_a$: LJ-potential with the harmonic oscillator:

$$U_b = \sum_{i \neq j} U_{\text{LJ}}(\|X_i - X_j\|) + \frac{1}{2} \sum_{n=1}^{N} \left\| X_n - \frac{1}{N} \sum_{n'=1}^{N} X'_n \right\|^2 \tag{158}$$

where

$$U_{\text{LJ}}(r) = 4\epsilon \left[ \left( \frac{\sigma}{r} \right)^{12} - \left( \frac{\sigma}{r} \right)^6 \right] \tag{159}$$

In our experiments, we set $\sigma = \epsilon = 1$.

- $S_b$: the harmonic oscillator:

$$U_a = \frac{1}{2} \sum_{n=1}^{N} \left\| X_n - \frac{1}{N} \sum_{n'=1}^{N} X'_n \right\|^2 \tag{160}$$

where $X_n \in \mathbb{R}^3$ is the coordinate for $n$-th particle in the system.

### F.1.3  Alanine dipeptide - solvation (ALDP-S)

We consider the solvation free energy between ALDP in the vacuum environment and with implicit solvent, defined with AMBER ff96 classical force field. Specifically, the samples were gathered from a 5 microsecond simulation under 300K with Generalized Born implicit solvent implemented in `openmmtools` Chodera et al. [2025]. The Langevin middle integrator implemented in Eastman et al. [2023] with a friction of 1/picosecond and a step size of 2 femtoseconds was used to harvest a total of 250,000 samples. The same sampling protocol was used in the following paragraph as well. Below are settings for $S_a$ and $S_b$:

- $S_a$: ALDP in the vacuum environment;
- $S_b$: ALDP in implicit solvent.

We also rescale each target scale by 20, i.e., we define the energy as $U\left(\frac{x}{20}\right)$. Note that this will only change the scale of input and the score, with no influence on the free energy difference as long as we apply the same scaling to both targets.

### F.1.4  Alanine dipeptide - transition (ALDP-T)

We consider Alanine dipeptide in the vacuum. As shown in Figure 2a, there are two metastable states in this system. We therefore consider the transition free energy between them. Below are settings for $S_a$ and $S_b$:

- $S_a$: ALDP in the vacuum environment, $\phi \in (0, 2.15)$;
- $S_b$: ALDP in the vacuum environment, $\phi \in [-\pi, 0] \cup [0, \pi)$.

Similar to the solvation case, we rescale each target scale by 20.

### F.1.5  $\varphi^4$ lattice field theory

For $\varphi^4$ experiments, we consider reweighting the histograms obtained from two umbrella samplings with the free energy estimated by our approach. The random variables here are field configurations $\varphi \in \mathbb{R}^{L \times L}$, and the energy function is defined as

$$U(\varphi) = \sum_x \left( -2 \sum_\mu \varphi_x \varphi_{x+\mu} + (4 + m^2)\varphi_x^2 + \lambda \varphi_x^4 \right) \tag{161}$$

Here, we use $\varphi_x$ to represent the value of $\varphi$ at index $x$. We choose $m^2 = -1, \lambda = 0.8$ following Albergo and Vanden-Eijnden [2024]. The two states are defined with two different umbrella samplings:

- $S_a$: $U_a(\varphi) = U(\varphi) + \frac{k_1}{2}(\frac{1}{L^2} \sum_x \varphi_x - \mu_1)^2$;
- $S_b$: $U_b(\varphi) = U(\varphi) + \frac{k_2}{2}(\frac{1}{L^2} \sum_x \varphi_x - \mu_2)^2$.

where $k_1 = k_2 = 10$, and $\mu_1 = -0.3, \mu_2 = 0.6$. We deliberately choose asymmetric values, as a symmetric setup will render the analysis less complicated.

We then estimate the free energy difference $\Delta F = F_b - F_a$ with our proposed approach. With this estimate, we construct the histogram of the average magnetization by reweighting the samples from $U_a$ and $U_b$. Concretely, for each bin $\xi$ in the histogram, we compute its reweighted probability as

$$P(\xi) \propto \frac{n_a(\xi) + n_b(\xi)}{N_a \exp(F_a - \frac{k_1}{2}(\xi - \mu_1)^2) + N_b \exp(F_b - \frac{k_2}{2}(\xi - \mu_2)^2)} \tag{162}$$

where $N_a$ denotes the total number of samples from $u_a$, and $n_a$ is the number of those samples falling in bin $\xi$.

## F.2 Hyperparameter

We include hyperparameters for model training and evaluation in Table 8. We explain some hyperparameters in the following:

- $\alpha_t, \beta_t, \gamma_t$: these parameters define the interpolant: $X_t = \alpha_t X_0 + \beta_t X_1 + \gamma_t \epsilon$, where $\epsilon \sim \mathcal{N}(0, I)$;
- OT pair: To facilitate training, instead of randomly sampling a pair $(X_0, X_1)$, we compute an optimal transport (OT) plan to select pairs of data points that are closer to each other. We use the implementation by Tong et al. [2024] (MIT License) to find the OT pair.

  Additionally, we note that standard OT chooses the closest pair in the Euclidean distance, which does not directly apply to our rotation-invariant alanine dipeptide and permutation/rotation-invariant Lennard-Jones data. To solve this problem, we first *canonicalize* all data points. Specifically, we select one sample as the reference system and rotate all other samples from both states to align with this reference using the Kabsch algorithm [Kabsch, 1976]. For the Lennard-Jones system, we additionally canonicalize atom permutations by applying the Hungarian algorithm [Kuhn, 1955, 1956] to find the optimal assignment that minimizes the distance matrix between each sample and the reference. Similar approaches were also adopted by Klein et al. [2023]. In practice, we found this significantly accelerates the training and enhances the performance for larger systems.

- OT batch size: instead of running OT on the entire dataset, we run OT within a much smaller batch to ensure a low running cost. Note that this number is different from the "batch size".
- FM warm up: we found it is helpful to warm up the vector field network with flow matching, especially for GMM in high-dimensional space. If we use this warm-up, we will put the iteration number in the table; otherwise, we will leave "-".
- $\sigma_t$: recall that during simulation, we run the forward and backward SDEs as defined in Equations (23) and (24). $\sigma_t$ is the diffusion term in these SDEs. We note that $\sigma_t$ is not the noise level for the stochastic interpolants (which is $\gamma_t$).

Table 8: Hyperparameters of our experiments.

| Hyperparameters | GMM | | LJ | | | ALDP-S/T |
|---|---|---|---|---|---|---|
| | $d=40$ | $d=100$ | $d=55\times3$ | $d=79\times3$ | $d=128\times3$ | $d=22\times3$ |
| **Model and Interpolant choices** | | | | | | |
| Network architecture | MLP | | SE(3)-GNN | | | |
| Network size | 5, 400 | | 4, 64 | | | |
| $\alpha_t$ | $1-t$ | | | | | |
| $\beta_t$ | $t$ | | | | | |
| $\gamma_t$ | $\sqrt{at(1-t)}, a=0.05$ | | | | | |
| **Training** | | | | | | |
| learning rate | 0.001 | | | | | |
| batch size | 1,000 | 1,000 | 100 | 30 | 20 | 500 |
| iteration number | 50,000 | 200,000 | 10,000 | 20,000 | 40,000 | 20,000 |
| OT pair | No | Yes | No | Yes | Yes | Yes |
| OT batch size | - | 1,000 | - | 500 | 500 | 500 |
| FM warm up | - | 50,000 | - | - | - | - |
| **Simulation and Estimation** | | | | | | |
| number of discretization steps | 500 | | | | | |
| evaluating sample size | 1,000 | 5,000 | 1,000 | 1,000 | 1,000 | 1,000 |
| $\epsilon$ | 0.01 | | | | | |

## F.3 Hyperparameters and Settings for One-sided FEAT

**Network.** For one-sided FEAT, we only need one network to parameterize the score/mean/noise for the diffusion process. We adopt the parameterization (precisely, $c_{in}, c_{skip}, c_{out}$) following Karras et al. [2022], with an EGNN (fully-connected) as the network to prediction the mean $\mathbb{E}[X_0|X_t]$. We increase the EGNN size with 5 hidden layers, each with 256 hidden units.

**Training details.** We train the network for 200,000 iterations, with a batch size of 20 for both Chignolin and Ala-4. We observe a better convergence for larger batch size, while we choose 20 to fit in our GPU. We keep an EMA with rate 0.99.

**Estimation details.** We estimate the free energy for each states with 500 discretization steps. We use 1,000 samples for ALA-4 and 3,000 samples for Chignolin. Additionally, we found it is more stable to use DDPM kernel [Ho et al., 2020] as the denoising and noising kernel instead of EM discretization kernel. This will only slightly change the formulation of the Gaussian expression, with other key components of FEAT unchanged.

Precisely, the forward SDE is defined following Karras et al. [2022] as:

$$\mathrm{d}X_t = \sqrt{2t}\overrightarrow{\mathrm{d}B_t}, X_0 \sim \mu \tag{163}$$

where $\mu$ is the target system distribution. The backward SDE is defined as:

$$\mathrm{d}X_t = 2t\nabla U_t^\theta(X_t)\mathrm{d}t + \sqrt{2t}\overleftarrow{\mathrm{d}B_t}, X_T \sim \mathcal{N}(0, T^2 I) \tag{164}$$

where $\nabla U_t^\theta$ is the learned score network. The score and the mean-prediction $\mathbb{E}[X_0|X_t]$ are connected with Tweedie's formula:

$$-\nabla U_t(X_t) = -\frac{X_t - \mathbb{E}[X_0|X_t]}{t^2} \tag{165}$$

The DDPM kernels for this pair of SDEs are:

$$\mathcal{N}^+(X_{t_{i+1}}|X_{t_i}) = \mathcal{N}\left(X_{t_{i+1}}|X_{t_i}, (t_{i+1}^2 - t_i^2)I\right) \tag{166}$$

$$\mathcal{N}^-(X_{t_i}|X_{t_{i+1}}) = \mathcal{N}\left(X_{t_i}|\frac{t_i^2}{t_{i+1}^2}X_{t_{i+1}} + (1 - \frac{t_i^2}{t_{i+1}^2})\mathbb{E}[X_0|X_{t_{i+1}}], \frac{t_i^2}{t_{i+1}^2}(t_{i+1}^2 - t_i^2)I\right) \tag{167}$$

### F.4 Computing Resources

All experiments are run on a single 80G NVIDIA H100.

### F.5 Baseline and Reference Settings

**Target FEP.** We use the same parameter as Table 8. Note that the interpolant becomes $X_t = \alpha_t X_0 + \beta_t X_1$, and the simulation process will be ODE. We do not align the iteration number to be the same as SI. Instead, we run the training until convergence.

**Neural TI.** We use the same parameter as Table 8. We parameterize the energy network instead of the score network in order to use the TI formula. Specifically, we take the output of the network, and take an inner product with the input to form a scalar as the energy.

**Neural TI Preconditioning Design.** We include preconditioning for the energy network to ensure boundary conditions and also to increase the accuracy of the learned energy $U_t$.

For GMM, we set the energy network to be

$$U(x_t, t) = a_t U_A(x_t) + b_t U_B(x_t) + c_t U_\theta(x_t, t) \tag{168}$$

where $a_t = \exp[f_\theta(t) - f_\theta(0)] \cdot (1-t)$, $b_t = \exp[g_\theta(1) - g_\theta(t)] \cdot t$ and $c_t = \exp[h_\theta(t)] \cdot t \cdot (1-t)$. $f_\theta, g_\theta, h_\theta, U_\theta$ are neural networks. We can exactly ensure the boundary condition by this.

For LJ, $U_A$ and $U_B$ is more sensitive to noisy $x_t$. Therefore, inspired by Máté et al. [2024b], we set the energy network to be

$$U(x_t, t) = b_t \cdot U_{\mathrm{LJ}}(x_t, r_t, a_t) + c_t \cdot U_\theta(x_t, t)$$

where $r_t$ is the radius parameter in LJ, ranging from 0 to 1. $a_t = \exp(\alpha_\theta(t)) \cdot t \cdot (1-t)$ is a smooth parameter, as used by Máté et al. [2024b]. $b_t = 1 - \exp(\beta_\theta(t)) \cdot t \cdot (1-t)$ and $c_t = \exp(\gamma_\theta(t)) \cdot t \cdot (1-t)$ are scalar to ensure boundary conditions.

Due to this specific choice of smoothing parameters, we failed to design a stable preconditioning for ALDP and hence did not compare FEAT with neural TI on ALDP.

**MBAR.** We use MBAR (with `pymbar`) to obtain the reference value for the LJ system and ALDP:

- LJ: for LJ-potential, we create $N$ distributions as follows:

$$U_i = \sum_{i \neq j} U_{\mathrm{LJ},i}(\|X_i - X_j\|) + \frac{1}{2} \sum_{n=1}^{N} \left\| X_n - \frac{1}{N} \sum_{n'=1}^{N} X'_n \right\|^2 \tag{169}$$

where

$$U_{\mathrm{LJ},i}(r) = 4\epsilon_i \left[ \left(\frac{\sigma_i}{r}\right)^{12} - \left(\frac{\sigma_i}{r}\right)^6 \right] \tag{170}$$

We set $\sigma_1 = \epsilon_1 = 1$, $\sigma_N = \epsilon_N = 0$, and let $\sigma_n = \epsilon_n$ vary as a linear interpolant between 1 and 0 For LJ-55/79, we use $N = 41$ distributions, and for LJ-128, we choose $N = 81$. We run the Metropolis-adjusted Langevin algorithm (MALA) to obtain 20,000 samples for each marginal, and then randomly choose 1,000 of them for each marginal to decorrelate the samples.

- ALDP-S: We set 11 distributions by modulating the solvation parameter with a factor $\lambda \in [0,1]$ that scales the charges in the 'CustomGBForce' in the force field with OpenMM. Specifically, we modify the force by the following code:

```
for force in system.getForces():
    if force.__class__.__name__ == 'CustomGBForce':
        for idx in range(force.getNumParticles()):
            charge, sigma, epsilon = force.getParticleParameters(idx)
            force.setParticleParameters(idx, (charge*lamb, sigma, epsilon))
```

Listing 1: Python code example for changing the solvation strength.

The 11 distributions are set with $\lambda = 0.0, 0.1, \ldots, 1.0$. We run MD for each distributions using the setup described in Appendix F.1.3, and randomly choose 2,000 of them for each marginal to decorrelate the samples to run MBAR.

- ALDP-T: we define three distinct distributions: (1) a distribution containing only $S_a$, where the energy in regions corresponding to state $S_b$ are set to $+\infty$, (2) a distribution containing only $S_b$, where the energy in regions corresponding to state $S_a$ are set to $+\infty$; and (3) a full distribution that includes both $S_a$ and $S_b$.

