# OpenReview forum: "FEAT: Free energy Estimators with Adaptive Transport"
_NeurIPS.cc/2025/Conference — NeurIPS 2025 poster_

### Official Review · Reviewer_HXX2 · 2025-06-13

**Clarity:** 3
**Significance:** 3
**Originality:** 3
**Rating:** 4
**Confidence:** 3

**Summary:**

FEAT introduces a learned stochastic‐dynamics framework for free‐energy estimation that adapts transport maps via time‑dependent vector fields and score functions. It unifies Jarzynski’s equality and Bennett’s acceptance‐ratio into two asymptotically unbiased, minimum‑variance estimators (FEAT‑J and FEAT‑BAR). By training these flows to minimize dissipation, FEAT reduces estimator variance without requiring intermediate sampling.

**Questions:**

In Table 3:
Row about parameter size: The network sizes are listed as "5, 400" and "4, 64."
What exactly do these numbers refer to? Are they the number of layers and hidden units, or do they represent the total number of learnable parameters?
If they do correspond to a small number of parameters, can you justify the use of an H100 GPU?

Throughout the dataset description, I did not find any information about train/test splitting.
Could you clarify how the data was handled during training and validation?
Was any explicit split used, or is this more of an overfitting-style exercise without separate validation?

**Ethical Concerns:**

["NO or VERY MINOR ethics concerns only"]

**Final Justification:**

After the rebuttal phase, the authors addressed my concerns by changing how some experiments were conducted. With the new results, I feel more inclined to accept the paper

**Limitations:**

yes

**Paper Formatting Concerns:**

-

**Quality:**

3

**Strengths And Weaknesses:**

Strength

Strong theoretical foundation, supported by extensive proofs.

Weaknesses

In Table 1 results for Neural TI perform poorly: although the text offers some justification, something should be done to make these results a fair comparison, otherwise these results do not have any significant meaning.

For certain systems (specifically ALDP‑S and ALDP‑T), the standard deviations are so large that comparing only the means does not ensure these estimates are trustworthy or reproducible under slightly different setups.

The lack of available code enhances these concerns, as it prevents independent verification and deeper investigation into the above issues.

---

> ### Author Rebuttal · Authors · 2025-07-31
>
> Thank you for your review. We have addressed your concerns with improved baseline comparisons, detailed statistical analysis, experimental details, and code availability. We are happy to answer any further questions you might have. Should you find our reply satisfactory, we kindly invite you to reconsider your rating.
>
> ## Responses to Weaknesses
>
> > ### 1. In Table 1 results for Neural TI perform poorly: although the text offers some justification, something should be done to make these results a fair comparison, otherwise these results do not have any significant meaning.
>
> Thank you for your suggestion. In our experiment, we tried to keep the network architecture as consistent as possible across different approaches. However, we fully agree that this is not ideal for neural TI, which benefits from task-specific architectural design to ensure boundary conditions and also mitigate the "blindless issue" of score matching losses (e.g., score is insensitive to error in mode weights when modes are separate).
>
> Following your suggestion, we **re-evaluated neural TI on GMM-40 and LJ using a better-designed network architecture** in the following.
>
> #### **Updated network design for neural TI:**
>
> - for GMM, we set the energy network to be
>
> $
> U(x_t, t) =  a_t U_A(x_t) + b_t U_B(x_t) + c_t U_\theta (x_t, t)
> $
>
> where $a_t = \exp[f_\theta(t) - f_\theta(0)] \cdot (1-t)$, $b_t = \exp[g_\theta(1) - g_\theta(t)] \cdot t$ and $c_t = \exp(h_\theta(t))  \cdot t \cdot (1-t)$, where $f_\theta, g_\theta, h_\theta, U_\theta$ are neural nets and by which we can exactly ensure the boundary condition.
>
> - for LJ, $U_A$ and $U_B$ are more sensitive to noisy $x_t$. Therefore, inspired by [1],  we set the energy network to be
>
> $
> U(x_t, t) = b_t \cdot U_\text{LJ}(x_t, r_t, a_t) + c_t \cdot U_\theta(x_t, t)
> $
>
> where $r_t$ is the radius parameter in LJ, ranging from  0 to 1. $a_t = \exp(\alpha_\theta(t) )\cdot t \cdot (1-t)$ is a smooth parameter inspired by [1]. $b_t = 1 - \exp(\beta_\theta(t))\cdot t\cdot (1-t)$ and $c_t = \exp(\gamma_\theta(t)) \cdot t \cdot (1-t)$ are scalar to ensure boundary conditions.
>
>
> #### **Updated neural TI results:**
>
> With these updated architectures, we obtained the following **updated results for neural TI**:
>
> - **GMM-40: 0.366** (reference value: 0)
> - **LJ-55: 235.894** (reference value: 234.77)
> - **LJ-79: 354.8852** (vs. reference 357.43)
>
> These results show that neural TI can achieve competitive performance when supported by carefully designed network architectures.
> We will provide updated results on other targets and provide error bars, and we will ensure a fairer discussion on neural TI  in our final version.
>
> [1] Máté, B., Fleuret, F. and Bereau, T., 2025. Solvation free energies from neural thermodynamic integration. The Journal of Chemical Physics, 162(12).
>
>
> > ### 2. For certain systems (specifically ALDP), the standard deviations are so large that comparing only the means does not ensure these estimates are trustworthy or reproducible.
>
> We completely agree that both mean and variance are crucial for assessing method reliability. We have conducted extensive additional experiments to address this concern.
>
> We performed **12 runs** (6 different model seeds × 2 estimates per model) and provide more statistics:
>
> | Method | All values                                                                                      | Mean        | Std Dev     |
> |--------|-------------|-------------|-------------|
> | FEAT   | 29.43451, 29.49945, 29.37685, 29.40085, 29.32738, 29.37606, 29.50049, 29.44383, 29.34062, 29.32738, 29.50566, 29.43218 | 29.41 | 0.06 |
> | Target FEP   | 29.26557, 29.68326, 29.16319, 29.59889, 29.36639, 29.50764, 29.59858, 29.52172, 29.34441, 29.47849, 29.35864, 29.34831 | 29.44  | 0.15 |
>
> Their 25, 50 and 75-quantiles are:
>
> - FEAT: 29.367, 29.417, 29.458
> - Target FEP: 29.347, 29.422, 29.541
>
> We can observe that FEAT gives more stable results when taking standard deviation and quantiles into account.
>
> > ### 3. Code availability
>
> We totally understand your concern. Due to this year’s policy, we are not allowed to directly provide you with links pointing to our code; therefore, **we have sent our code link to AC**.
>
> Additionally, we will make our code publicly available in the next version of our paper.
>
> ## Responses to Questions
>
> > ### Q1: Network architecture specifications in Table 3
>
> Sorry for the confusion. "5, 400" refers to 5 layers of neural networks with 400 hidden dimensions; similarly, "4, 64" refers to 4 layers of neural networks with 64 hidden dimensions.
>
> Our method does not necessarily require an H100 to train, smaller GPUs also suffice.  We further report the GPU usage of our targets:
>
> | target | Batch size | GPU usage |
> |---|---|---|
> | LJ-55 | 100 | ~28 G |
> | LJ-79 | 30 | ~18 G |
> | LJ-128 | 20 | ~19 G |
> | GMM-40 | 1000 | ~1G |
> | GMM-100 | 1000 | ~2G |
> | ALDP | 500 | ~20G |
>
> > ### Q2: Train/test splitting and validation procedures
>
>
>
> We address this from two perspectives:
>
> **a) Train/test split:**
> For a given pair of states A & B, we aim to fit a transport between A and B as much as possible, and in theory, we assume that we can sample sufficient configurations in these states.
> From this perspective, our approach is more akin to stochastic approximation methods from Robbins-Monro [2] and therefore has minimal overfitting issues.
>
> On the other hand, when the data set is provided beforehand and kept fixed, it may not be representative enough, and using the same set for training and testing may introduce bias.
> To this end, we provide an 8:2 train/test split result on ALDP.
> The result is not significantly better than that reported in our manuscript, indicating that the dataset we used is sufficient.
>
> | Target | Result | Ground Truth |
> |--------|--------|--------------|
> | ALDP-S | 29.45 ± 0.06 | 29.43 ± 0.01 |
> | ALDP-T | -4.39 ± 0.09 | -4.25 ± 0.05 |
>
> **b) Hyperparameter tuning:**
> Another question is whether a train/validation split is needed for hyperparameter tuning.
> The answer is No: in our approach, we tune all hyperparameters based on transport quality by monitoring the energy of transported training samples and the gap between forward and backwards estimators.
>
> These validation metrics eliminate the need for separate validation sets.
>
> [2] Robbins, H. and Monro, S., 1951. A stochastic approximation method. The annals of mathematical statistics, pp.400-407.

---

> > ### Comment · Reviewer_HXX2 · 2025-08-04
> >
> > Thank you very much for your detailed answer! Could you also elaborate on the standard deviation values for your updated Neural TI model as the previous standard deviations for that model were also raising concerns for me

---

> > > ### Author Response · Authors · 2025-08-04
> > >
> > > Thank you for your further suggestion! We run 3 times for GMM and LJ-79. Both achieve reasonable values for mean and std:
> > >
> > > - GMM-40:  0.11 +- 0.20 (vs. reference value: 0)
> > > - LJ-79: 356.9 +- 1.8 (vs. reference 357.43)
> > >
> > > We will report error bars across more runs and provide updated results on other targets as well in our final version, and ensure a fair discussion on neural TI. Please let us know if there are any other suggestions, and thanks again for helping us strengthen our manuscript.

---

> > > > ### Comment · Reviewer_HXX2 · 2025-08-05
> > > >
> > > > I appreciate the effort you put into addressing my concerns. I am happy to reevaluate my score in light of your explanations

---

> > > > > ### Author Response · Authors · 2025-08-07
> > > > >
> > > > > Dear reviewer,
> > > > >
> > > > > We are glad our rebuttal has addressed your concerns and thank you for engaging in the rebuttal and helping us strengthen our paper!
> > > > >
> > > > > Best Regards,
> > > > > Authors

---

### Official Review · Reviewer_Enj8 · 2025-06-19

**Clarity:** 4
**Significance:** 4
**Originality:** 4
**Rating:** 5
**Confidence:** 4

**Summary:**

This work presents FEAT, a novel framework for free energy calculation that provides consistent, minimum-variance estimators based on escorted Jarzynski equality and controlled Crooks theorem. The framework leverages stochastic interpolant to learn transports from two endpoint systems directly, with no necessity for intermediate states. It has been proved that the estimator is asympototically unbiased even with imperfect learned potentials and discretization errors. Experiments on Gaussian mixtures, LJ and alanine-dipeptide show promising improvements on free energy estimations compared to all baselines.

**Questions:**

1. From my perspective, since the proposed method relies entirely on neural networks and potential energy calculations for free energy estimation without requiring MD simulations, it should be applicable to larger molecular systems in principle. Could the authors provide experimental results on more complex molecules like proteins, along with an analysis of the computational efficiency?
2. When training the stochastic interpolant, each iteration requires selecting a data pair $(x_a,x_b)$. In the case of FEAT, are the data points independently sampled from $\mu_a$ and $\mu_b$ during training, or are the datasets $x_a^{1:N}$ and $x_b^{1:N}$ pre-paired in a one-to-one fashion? A brief explanation of whether these two approaches would lead to different training outcomes would be appreciated.
3. There appears to be a typo in Figures 2(b) and 2(c): the references should be to Eq. (27) and Eq. (30), rather than Eq. (24)/(25) and Eq. (27).

**Ethical Concerns:**

["NO or VERY MINOR ethics concerns only"]

**Final Justification:**

My main concern was whether FEAT can scale to larger molecular systems. In the rebuttal, the authors provided additional experiments on alanine dipeptide and Chignolin, where FEAT demonstrated promising results. I believe this supports the practical applicability of FEAT to larger systems. Therefore, I will maintain my score.

**Limitations:**

It is strongly recommended that FEAT be evaluated on larger molecular systems to demonstrate its applicability to more complex, real-world scenarios.

**Quality:**

4

**Strengths And Weaknesses:**

**Strengths**
- The paper is exceptionally well-organized and clearly articulated, making it easy to follow even for readers without a background in computational physics.
- The proposed method demonstrates strong originality and practical relevance, and is supported by rigorous theoretical justifications.
- The proposed method enables the estimation of free energy without requiring access to intermediate states, and it is theoretically shown to remain a consistent estimator even when the learned score is imperfect on the boundaries. This significantly reduces the reliance on computational resources and model parametrization.

**Weaknesses**
- The proposed method estimates free energy without relying on MD simulations and is, in principle, fully extendable to macromolecular systems. However, it is regrettable that the experiments are limited to small molecules (e.g., alanine dipeptide), without demonstrating its applicability to larger systems.

---

> ### Author Rebuttal · Authors · 2025-07-30
>
> Thank you for your detailed and insightful review. We are delighted that you found our work exceptionally well-organized, original, effective, and significant. We now address your concerns with new experiments on larger molecular systems and clarifications on experimental details:
>
> ## Responses to Weaknesses and Questions
>
> > ### Q1 & Weakness: Experimental results on more complex molecules
>
> Thank you for your great suggestions. Following your recommendation, we now provide results on significantly larger systems:
>
> **Alanine Tetrapeptide Results:**
> - System size: 43 atoms (129 dimensions)
> - FEAT estimate: **65.51 ± 1.51 kcal/mol**
> - Reference value (MBAR with 20 intermediate distributions): **64.09 kcal/mol**
>
> **Chignolin Results:**
> - System size: 175 atoms (525 dimensions)
> - For this larger system, we found that it is more effective to decompose the estimation task: instead of directly learning a transport map between the vacuum and solvent states, we estimate the free energy using FEAT in two stages—from a Gaussian to vacuum, and from a Gaussian to solvent—and then take the difference between the two estimates.
> - FEAT estimate: **190.784  +- 0.414 kcal/mol** (due to time limit, we did not train 3 times, and this std is obtained by doing 3 inferences with one model.)
> - Reference value (MBAR with 20 intermediate distributions): **190.887 kcal/mol**
>
> These results demonstrate FEAT's applicability to larger, more realistic systems. We will update these results in our camera-ready version.
>
> > ### Q2: Data pair in stochastic interpolant training
>
> This is an excellent question.
> The FEAT framework is general and supports multiple pairing strategies, for example:
> - OT: Using optimal transport (OT) plans for structured correspondence
> - Random: Independent sampling from each endpoint distribution
>
> In our experiments on simpler systems such as GMM-40 and LJ-55, we use randomly selected pairs. For more complex targets, we sample batches randomly but apply mini-batch OT to determine better pairings, which helps to stabilize training and improve network fitting. We found that omitting the OT step leads to slower convergence and reduced performance, as random pairing can lead to less "straight" vector fields, making training more challenging.
>
> In table 3, we have a hyperparameter called "OT pair" to reflect if we use this approach in our training. We will provide a brief discription and discussion in our camera-ready version.
>
>
> > ### Q3: Figure 2 equation reference typo
>
> Thank you for catching it. We will fix in our camera-ready version.

---

> > ### Comment · Reviewer_Enj8 · 2025-08-01
> >
> > I appreciate the authors’ additional experiments on alanine dipeptide and Chignolin. The performance of FEAT on larger systems is also very promising. I will maintain my score. Wish you all the best!

---

> > > ### Author Response · Authors · 2025-08-07
> > >
> > > Dear reviewer,
> > >
> > > We are glad our rebuttal has addressed your concerns and thank you for engaging in the rebuttal and helping us strengthen our paper!
> > >
> > > Best Regards,
> > > Authors

---

### Official Review · Reviewer_7RQk · 2025-07-02

**Clarity:** 3
**Significance:** 2
**Originality:** 2
**Rating:** 5
**Confidence:** 2

**Summary:**

The paper introduces Free Energy Estimators with Adaptive Transport (FEAT), a method for estimating free energy. It requires only samples from the endpoint distributions and learns a continuous transport map between two endpoint distributions, $S_A$​ and $S_B$​, using stochastic interpolants. Bi-directional trajectories that carry samples between $S_A$​ and $S_B$​ are simulated using time-dependent drift and gradient-energy models, parameterized as neural networks. These samples are then used to estimate the free energy. From my perspective, the key novelty is that free energy is not estimated via importance sampling, but instead through a minimum-variance estimator, such as the Bennett Acceptance Ratio (BAR) method. In experiments involving Gaussian Mixture Models, Lennard-Jones particles, and alanine dipeptide systems, FEAT achieves results that are closer to reference values than baseline methods based on target free-energy perturbation approach using the flow-matching and thermodynamic integration.

**Questions:**

Questions:
1. This work assumes that the density is the Boltzmann weight $\exp[-u(x)]$. Can FEAT be generalized to non-Boltzmann sampling?
2. Theoretical assumptions. When deriving eq. 36, do you assume perfect control, such that samples follow $u_t$ precisely? As I understand it, you assume that forward and backward models are close. What happens if they diverge, e.g., due to numerical errors?
3. It is unclear how you estimate $\dot{I}_t$ for learning $v_t$ (eq. 19). I assume the estimate could be noisy. Did you apply any smoothing techniques? If so, which ones?
4. In Figure 2 (b) and (c), the convergence rate of FEAT is compared with estimates using only forward or backward simulations (eq. 27). To me, it appears that averaging the red curves (eq. 27) would yield a convergence rate similar to the blue curve (eq. 30), which would make for a fairer comparison. Also, the equation numbers in the legends of subplots (b) and (c) seem incorrect.
5. There is inconsistent notation between the main text and Algorithm 1. In the text, $v$ denotes the vector field, while in Algorithm 1 it is denoted as $u$.
6. In the Limitations, it is mentioned that FEAT uses only samples from the endpoints. However, if intermediate distributions between the endpoints are available, can FEAT make use of them?
7. Regarding the training of neural networks: how do you control overfitting? What is the train/validation split ratio?
8. For completeness, it would be helpful to add in Table 2 the estimated time required for reference value computation. This would highlight the cost of sampling from intermediate distributions and using MBAR.
9. Typo: Line 185: Corollary 3.1 (Escorted Jarzynski **Equality** ...
10. How does the number of discretization steps affect FEAT and the baselines? Is FEAT more robust than the baselines in this regard? How does the number of steps required for good performance scale with system size? How does it depend on the similarity between endpoint distributions (e.g., as measured by KL divergence)?
11. Similar to Question 10, but regarding sample size instead of discretization steps.
12. Regarding Equations 21 and 22: You use the target score matching approach from [2]. I reviewed that paper, and the choice of intervals $\mathcal{U}(0, 0.5)$ and $\mathcal{U}(0.5, 1)$ appears arbitrary. You report results for the same intervals — did you try other options, such as $[0, 0.25]$ and $[0.75, 1]$? Alternatively, could you provide a rationale for the chosen intervals?
13. Question about generalization: After training, can FEAT be applied to different systems without changing the neural network architecture or retraining? If yes, results from such experiments would be valuable.
14. What is the performance degradation when omitting the data pre-processing step (canonicalization), both for FEAT and the baselines?


[2] B. Máté, F. Fleuret, and T. Bereau. Solvation free energies from neural thermodynamic integration. arXiv preprint arXiv:2410.15815, 2024b


I expect the authors to address Questions 1–9 in the rebuttal and to at least partially address some of the experiment-related questions (10–14).

**Ethical Concerns:**

["NO or VERY MINOR ethics concerns only"]

**Final Justification:**

The authors have addressed many of my theoretical and practical concerns, which were listed in the Weaknesses and Questions sections. However, the issue of selecting appropriate time intervals for target score matching remains underexplored, even if the default values "just work." The authors argue that this topic is tangential to their main contribution, but it is, in fact, an essential part of the proposed approach.
Overall, I am positive about the paper and the authors’ responses, so I have raised my score.

**Limitations:**

yes

**Paper Formatting Concerns:**

The main body of the paper exceeds 9 pages slightly.

**Quality:**

3

**Strengths And Weaknesses:**

Strengths:
1. The proposed approach is theoretically and technically sound. Limitations such as the requirement for access to samples from both systems are clearly stated by the authors.
2. Overall, the manuscript is well organized and easy to follow. In the appendix, the authors present an extended review of approaches to the problem of free-energy estimation, theoretical proofs, hyperparameters for FEAT and the baselines, as well as a comparison of inference time between FEAT and the baselines.
3. FEAT outperforms the baselines on the considered datasets, demonstrating its effectiveness in settings where samples from endpoint distributions are available. It is also computationally efficient compared to a baseline that achieves comparable quality.
4. The theoretical framework is based on the Escorted Jarzynski Equality, Crooks theorem, and Bennett acceptance ratio, and generalizes them to a novel use case involving learned non-equilibrium transport.

Weaknesses:
1. As there are no well-established state-of-the-art approaches or benchmarks for this task, the value of the work could be improved by evaluating additional baselines (e.g., those mentioned in related works, including DeepBar [1]) and by using datasets from those studies.
2. The paper could benefit from a more in-depth investigation into the robustness of FEAT (see questions for details). This would help practitioners understand where the method is most applicable.
3. The limitations section and the algorithm description contain essential information and should be moved from the appendix to the main part of the paper.
4. There are some typos and inconsistencies in notation throughout the text. Additionally, a few technical details remain unclear (see questions).


[1] Ding,  Xinqiang and Zhang,  Bin. DeepBAR: A Fast and Exact Method for Binding Free Energy Computation. The Journal of Physical Chemistry Letters, 2021.

---

> ### Author Rebuttal · Authors · 2025-07-31
>
> Thank you for your constructive review. We are pleased that you found our work theoretically sound, well-organized, effective, and novel. We have addressed your concerns with further experiments and clarifications. We are committed to addressing any further concerns you may have. Should you find our reply satisfactory, we kindly invite you to reconsider your rating.
> ## Responses to Weaknesses
>
> > ### W1: Comparison with DeepBAR as an additional baseline
>
> Thank you for this excellent suggestion!
> However, **FEAT is orthogonal to DeepBAR rather than directly comparable.**
>
> DeepBAR estimates free energy differences by first estimating absolute free energies for each system using generative models (e.g., NF or CNF), then taking their difference. **FEAT can actually improve DeepBAR by providing better transport learning for the absolute free energies.**
>
> We provide results on GMM and ALDP-S using DeepBAR with CNF and FEAT below:
>
> || GMM-40D (GT: 0) | GMM-100D (GT: 0) | ALDP-S (reference value: 29.43) |
> |--|--|--|--|
> | DeepBAR-CNF  | -0.06  | 0.16  | 28.68 |
> | DeepBAR-FEAT | -0.02  | -0.20  | 29.38  |
>
> DeepBAR-FEAT outperforms DeepBAR-CNF, especially on ALDP-S. Interestingly, on GMM-100, both achieve better performances compared to standard Target FEP w. FM and FEAT, likely because learning transport from Gaussian to GMM is easier than directly learning between two complex GMMs.
>
> We will include this discussion and comparison in our camera-ready version.
>
> > ###  W2: Additional baselines from related works
>
> Since DeepBAR did not release their datasets, we evaluated on a comparable system: Chignolin (175 atoms, 525 dimensions) in vacuum vs. implicit solvent:
> - Reference value (MBAR): 190.887 kcal/mol
> - DeepBAR-FEAT result: **190.784  +- 0.414 kcal/mol** (due to time limit, we did not train 3 times, and this std is obtained by doing 3 inferences with one model.)
>
> This showcased our approach's scalability.
>
> ## Responses to Questions
>
> > ### Q1: Can FEAT be generalized to non-Boltzmann sampling?
>
> Yes. FEAT can handle any distribution admitting a differentiable density $p(x)$. The Boltzmann form is not required: our framework applies to any probability distribution satisfying these mild regularity conditions.
>
> > ### Q2: When deriving Eq. 36, do you assume perfect control?
>
> No. Equation 36 relies entirely on importance sampling (lines 619-624) and does not require perfect control. This is a key advantage of FEAT: intermediate samples need not remain at equilibrium.
>
> However, we do need the control to be "reasonable" with sufficient overlap between forward and backward processes. Poor control can increase estimation error, but this is easily detected through:
> - Quality of transported samples
> - Gap between ELBO and EUBO
> - Discrepancy between forward and backward estimators
>
> > ### Q3: How do you estimate $\dot I_t$  for learning $v_t$?
>
> This appears to be a misunderstanding. The derivative of the interpolant $\dot I_t$ can be calculated exactly from our Eq.17: it does not need to be estimated or learned. Instead, we use it to learn $v_t(x) = \mathbb{E}[\dot I_t| I_t = x]$ (Eq.18; Eq. 2.10 in [1] ), which is exactly the minimizer of the loss in Eq.19 and is not noisy.
>
> [1] Michael S. Albergo, et al. Stochastic Interpolants: A Unifying Framework for Flows and Diffusions
>
> > ### Q4: Comparison between averaged forward-backward estimators vs. min-variance estimator
>
> This is a great suggestion! In fact, **the minimum-variance estimator (blue curve) is theoretically guaranteed to perform no worse than averaged forward-backward estimators (red curves)**, as it's derived to minimize estimation error.
>
> To see this, we provide an
> **empirical verification on ALDP-S:**
> - After 2000 iters: Blue = 29.30 +- 0.36; Avg red = 28.86 +- 1.19
> - After 5000 iters: Blue = 29.54 +- 0.16; Avg red = 29.56 +- 0.73
>
> > ### Q6: Can FEAT utilize intermediate distributions if available?
>
> Yes. We can learn multiple FEATs in sequence: Given intermediate distributions $U_2...U_{N-1}$ between $A$ (energy $U_1$) and $B$ (energy $U_N$), we learn FEAT between each neighboring pair $U_i$ and $U_{i+1}$. The total $\Delta F$ is the sum of individual estimates. This approach may reduce individual network burden compared to learning a single $A\to B$ transport.
>
> > ###  Q7: How do you control overfitting?
>
> We answer this from two perspectives:
>
> **a) Train/test split:**
> For a given pair of state A & B, we aim to fit a transport between A and B as much as possible, and in theory we assume that we can sample sufficient configurations in both states. From this perspective, our approach is more akin to stochastic approximation methods from Robbins-Monro [2] and therefore has minimal overfitting issues.
>
> On the other hand, when the data set is provided beforehand and kept fixed, it may not be representative enough, and using the same set for training and testing may introduce bias. To this end, we provide an 8:2 train/test split result on ALDP.  The result is not significantly better than that reported in our manuscript, indicating that the dataset we used is sufficient.
>
> | Target | Result | Ground Truth |
> |----|---|-----|
> | ALDP-S | 29.45 ± 0.06 | 29.43 ± 0.01 |
> | ALDP-T | -4.39 ± 0.09 | -4.25 ± 0.05 |
>
> **b) Hyperparam tuning:**
> Another question is whether a train/valid split is needed for hyperparam tuning.
> The answer is No:
> in our approach, we tune all hyperparameters based on transport quality by monitoring the energy of transported training samples, or the gap between forward and backward estimators. These validation metrics eliminate the need for separate validation sets.
>
> [2] Robbins, H. and Monro, S., 1951. A stochastic approximation method. The annals of mathematical statistics, pp.400-407.
>
> > ### Q8: Reference computation times for Tab 2 to highlight the cost of sampling from intermediate distributions
>
> For ALDP, generating samples for each target requires ~1 day on our machine. Collecting all the targets for MBAR can hence take between 1-10 days, depending on whether the simulations are run in parallel.
>
> In contrast, our training process takes only 1-2 hours, which is significantly faster than sampling from all intermediate densities.
> We will include these rough numbers and the corresponding discussion in our camera-ready version.
>
> > ###  Q10 & Q11: Robustness w.r.t. discretization steps and sample size
>
> Following your suggestion, we conducted robustness analysis on GMM:
>
> **FEAT**
> | #step| sample size| GMM-40D | GMM-100D |
> |-|-|--|--|
> | 50 | 5k| -0.05 +- 0.33 | -4.04 +- 1.92  |
> | 100|5k| 0.12 +- 0.07  | -6.43 +- 1.72  |
> | 500 |5k| 0.00 +- 0.06  | -3.59 +- 1.06  |
> | | | |  |
> | 500|500  | 0.13 +- 0.09  | -5.56 +- 1.87  |
> | 500 |1k | -0.04 +- 0.05 | -6.46 +- 1.86  |
> | 500 |5k| 0.00 +- 0.06  | -3.59 +- 1.06  |
>
> **Target FEP**
> | #step| sample size| GMM-40D | GMM-100D |
> |-|-|--|--|
> | 50 | 5k| -0.98 +- 0.23 | -15.44 +- 6.02 |
> | 100|5k| -0.48 +- 0.17 | -12.10 +- 4.80 |
> | 500 |5k|-0.12 +- 0.08 | -13.72 +- 2.46 |
> | | | | |
> | 500|500  |  -0.16 +- 0.37 | -22.24 +- 3.43 |
> | 500 |1k |  0.02 +- 0.18  | -19.73 +- 2.13 |
> | 500 |5k|-0.12 +- 0.08 | -13.72 +- 2.46 |
>
> These results **highlight FEAT's greater robustness toward fewer steps/samples**, compared to Target FEP w flow matching. The results obtained with fewer steps (larger discretisation error) also reflect our discussion in lines 238-239.
>
> > ###  Q12: Target score matching with different time intervals
>
> Following your suggestions, we evaluate FEAT with $t \sim U(0, 0.25)$ and $U(0.75, 1)$ in TSM on ALDP-S.  It achieves $29.37 \pm 0.02$ across 3 independent runs.
> Compared to the choice with $U(0, 0.5)$ and $U(0.5, 1)$ ($29.38\pm 0.04$), we do not see obvious difference.
>
> This is because TSM helps the score learn better at boundary.  As long as we sample $t$ close to the boundary, TSM can provide useful signal.
> Also, different from neural TI that fully relies on the learned energy, FEAT rely on non-equilibrium transports with both the score and the vector field, leading to a higher robustness towards the TSM choice.
>
> > ###  Q13: Can FEAT be applied to different systems without retraining?
>
> Yes. If we train the network using datasets from multiple targets and condition the network on the target parameters, the resulting model becomes transferable, allowing us to apply FEAT to similar systems without retraining.
>
> We showcase this on GMM-40 with different scaling factor: concretely, we scale the state B with scaling $0.5, 0.7, 0.9, 1.1, 1.3, 1.5$ when training. Our network will also take the scaling as an input. After training, we evaluate $\Delta F$ on unseen scaling factors and report the error and std across 3 runs in the following table:
>
> | Scaling | 0.45 | 0.6 | 0.8 | 1.0 | 1.2 | 1.4 | 1.55 |
> |--|--|--|--|--|--|--|--|
> | Error +- std  | -3.52 +- 1.13 | -0.07 +- 0.05 | -0.02 +- 0.06 | -0.01 +- 0.06 | 0.01 +- 0.04 | 0.003 +- 0.04 | 0.03 +- 0.13 |
>
> These results show that the transferable FEAT model achieves good accuracy across a range of unseen targets. Also, as expected, interpolation yields better performance than extrapolation.
>
> > ### Q14: Performance without data preprocessing
>
> Omitting the data processing step significantly complicates training, resulting in larger bias and variance. Following your suggestion, we provide an ablation study on ALDP-S.  Without pre-processing:
> - FEAT: $37.40 \pm 0.38$
> - Target FEP w FM: $44.2 \pm 4.0$.
>
> Neither method attains a reasonable value, while FEAT exhibits a slightly better performance.
>
> However, we emphasize that this is not a weakness of our proposed framework. Rather, it reflects a general challenge in generative modeling. We also expect that using higher-capacity architectures (e.g., transformers) could help mitigate this issue.
>
> ## Formatting and typos (Weakness 3/4 and Q5/9):
>
> Thank you for pointing these typos out! We will fix them in our camera-ready version.
>
> Broader Impact is not counted to the page limit per instruction, so our main text is within 9 pages.

---

> > ### Comment · Reviewer_7RQk · 2025-08-04
> >
> > Thank you for addressing many of my concerns.
> > 1. I see that for the Chignolin system, you report results only for the two-stage FEAT. What are the results for the one-stage FEAT?
> > 2. You mention that performance does not change much when using target score matching with different time intervals. Could this robustness to the choice of time intervals be a property unique to relatively simple systems? For example, does this robustness hold for the Chignolin system?

---

> ### Author Response · Authors · 2025-08-04
>
> Thank you for your continued participation and thoughtful feedback.
> > I see that for the Chignolin system, you report results only for the two-stage FEAT. What are the results for the one-stage FEAT?
>
> We found learning a map from one complex system to another system is harder than mapping from a simple system to only one complex system, This requires longer time to converge and also potentially better network with higher capacity. In our attempts, we obtained 314.2 +- 26.9  ( = **186.9 +- 16.0 kcal/mol vs reference value 190.8 kcal/mol**) using one-stage FEAT (obtained by 3 runs with one model). It is not crazily off, but it is worse than two-stage FEAT. We will include a discussion in our final version.
>
> > You mention that performance does not change much when using target score matching with different time intervals. Could this robustness to the choice of time intervals be a property unique to relatively simple systems? For example, does this robustness hold for the Chignolin system?
>
> - In fact, we did not observe a performance difference in our case on both GMM and ALDP, which ablates two systems with fairly different complexities.
>  - Whilst we agree with the reviewer that more ablation of the Uniform interval is indeed interesting, we would like to iterate that the used settings, which were inherited from previous works, are not the primary focus/contribution of our work.
>  - The Chignolin experiment  is expensive to run, and we will not be able to thoroughly test this during the discussion period. For completeness, we are happy to attempt an ablation of this in the final version, as it may be useful for the wider community. This is, however, tangential to our contribution.
>
>  Thank you again for your time, and we are happy to address any further concerns you may have.

---

> > ### Comment · Reviewer_7RQk · 2025-08-06
> >
> > Thank you for the responses. I have raised my score.

---

> > > ### Author Response · Authors · 2025-08-07
> > >
> > > Dear reviewer,
> > >
> > > We are glad our rebuttal has addressed your concerns and thank you for engaging in the rebuttal and helping us strengthen our paper!
> > >
> > > Best Regards,
> > > Authors

---

### Official Review · Reviewer_ywwp · 2025-07-03

**Clarity:** 3
**Significance:** 2
**Originality:** 4
**Rating:** 5
**Confidence:** 3

**Summary:**

In this paper, the authors employed an adaptive transport method called FEAT to predict free energy profiles for chemical reactions. FEAT can handle both equilibrium and non-equilibrium samples. FEAT was also applied to umbrella sampling analysis, replacing the traditional method WHAM.

**Questions:**

1. Could the authors include more real-world case studies (beyond toy systems) for free energy prediction?
2. Could the authors provide details on the training sets used for the alanine dipeptide (2-state), Lennard-Jones (LJ), and ALDP-S systems? Additionally, please clarify the differences between the "2-state alanine dipeptide" and "ALDP-S" systems
3. What are the limitations regarding the applicability of the FEAT method to explicit solvent systems or larger (e.g., biomolecular) systems?

**Ethical Concerns:**

["NO or VERY MINOR ethics concerns only"]

**Final Justification:**

Authors' rebuttal has address all my concerns.

**Limitations:**

The scope of evaluation appears limited to small model systems, with no comparative benchmarks against some MD driven systems like drug-target systems.

**Paper Formatting Concerns:**

No formatting concerns.

**Quality:**

3

**Strengths And Weaknesses:**

**Strengths**:

1. This work presents an effective application of deep learning models for free energy calculation, a relatively novel area within AI4Science research.
2. The method demonstrates application in several examples with real-world scenarios.

**Weaknesses**:

See limitations

---

> ### Author Rebuttal · Authors · 2025-07-30
>
> Thank you for your valuable review. We are pleased that you found the method in our paper effective and novel. We have addressed your concerns with new experiments on larger systems and detailed clarifications about our experimental setup. We are committed to addressing any remaining concerns you may have. Should you find our reply satisfactory, we kindly invite you to reconsider your rating.
>
> ## Responses to Questions
>
> > ### Q1: Could the authors include more real-world case studies (beyond toy systems) for free energy prediction?
>
> Thank you for this important suggestion. We provide FEAT's results on two larger systems: **Alanine Tetrapeptide** (polypeptide with 4 amino acids) and **Chignolin** (protein with 10 amino acids).
>
> **Alanine Tetrapeptide Results:**
> - System size: 43 atoms (129 dimensions)
> - FEAT estimate: **65.51 ± 1.51 kcal/mol**
> - Reference value (MBAR with 20 intermediate distributions): **64.09 kcal/mol**
>
> **Chignolin Results:**
> - System size: 175 atoms (525 dimensions)
> - For this larger system, we found that it is more effective to decompose the estimation task: instead of directly learning a transport map between the vacuum and solvent states, we estimate the free energy using FEAT in two stages—from a Gaussian to vacuum, and from a Gaussian to solvent—and then take the difference between the two estimates.
> - FEAT estimate: **190.784  +- 0.414 kcal/mol** (due to time limit, we did not train 3 times, and this std is obtained by doing 3 inferences with one model.)
> - Reference value (MBAR with 20 intermediate distributions): **190.887 kcal/mol**
>
> These results demonstrate FEAT's applicability to larger, more realistic systems. We will update these results in our camera-ready version.
>
> > ### Q2: Could the authors provide details on the training sets used for the alanine dipeptide (2-state), Lennard-Jones (LJ), and ALDP-S systems?
>
> We included comprehensive details about each system in Appendix G.1, including state definitions and sample collection procedures. Could you elaborate on which specific details are missing? We will ensure any additional details you suggest are included in our camera-ready version.
>
> **Clarification on System Differences:**
>
> **Two-state alanine dipeptide (ALDP-T):**
> - We split the Boltzmann distribution for alanine dipeptide in vacuum into two metastable state distributions
> - State A: blue shaded region in Fig 2(a)
> - State B: pink shaded region in Fig 2(a)
> - Physical meaning: Free energy difference when alanine dipeptide transitions between metastable conformational states
>
> **ALDP-S (Solvation):**
> - State A: entire Boltzmann distribution for alanine dipeptide in vacuum
> - State B: entire Boltzmann distribution for alanine dipeptide in implicit solvent
> - Physical meaning: Solvation free energy difference between vacuum and implicit solvent environments
>
> > ### Q3: What are the limitations regarding the applicability of the FEAT method to explicit solvent systems or larger (e.g., biomolecular) systems?
>
> Explicit solvent has been a long-standing challenge for simulating biomolecules with machine learning forcefields as explicit solvent molecules are often too large to fit into GPU memory. It is a challenge of independent interest and is orthongoal to what we proposed in our framework. We hence only consider the implicit solvent in our study.

---

> > ### Comment · Reviewer_ywwp · 2025-08-03
> >
> > Authors' rebuttal has address all my concerns. I have raised my score to 5.

---

> > > ### Author Response · Authors · 2025-08-07
> > >
> > > Dear reviewer,
> > >
> > > We are glad our rebuttal has addressed your concerns and thank you for engaging in the rebuttal and helping us strengthen our paper!
> > >
> > > Best Regards,
> > > Authors

---

### Decision · Program_Chairs · 2025-09-17

**Decision:**

Accept (poster)

**Comment:**

The FEAT framework introduced in this paper estimates free energy using adaptive transport. It is well-grounded in theory, and the paper is well written, and demonstrates improvements over baselines. The new results added during rebuttal were well taken by the reviewers, and all of them recommend acceptance. I also vote to accept the paper as FEAT represents a timely contribution to AI for science.